# Sample Efficient Myopic Exploration Through Multitask Reinforcement Learning with Diverse Tasks

**Ziping Xu**[*,1]**, Zifan Xu**[2]**, Runxuan Jiang**[3]**, Peter Stone**[2,4] **& Ambuj Tewari**[3]
[1]Harvard University, [2]The University of Texas at Austin, [3]University of Michigan, [4]Sony AI
[*]This work was done while Z.X. was a Ph.D. student at the University of Michigan.

## Abstract

Multitask Reinforcement Learning (MTRL) approaches have gained increasing attention for its wide applications in many important Reinforcement Learning (RL) tasks. However, while recent advancements in MTRL theory have focused on the improved statistical efficiency by assuming a shared structure across tasks, exploration–a crucial aspect of RL–has been largely overlooked. This paper addresses this gap by showing that when an agent is trained on a sufficiently *diverse* set of tasks, a generic policy-sharing algorithm with myopic exploration design like $\epsilon$-greedy that are inefficient in general can be sample-efficient for MTRL. To the best of our knowledge, this is the first theoretical demonstration of the "exploration benefits" of MTRL. It may also shed light on the enigmatic success of the wide applications of myopic exploration in practice. To validate the role of diversity, we conduct experiments on synthetic robotic control environments, where the diverse task set aligns with the task selection by automatic curriculum learning, which is empirically shown to improve sample-efficiency.

## 1 Introduction

Reinforcement Learning often involves solving multitask problems. For instance, robotic control agents are trained to simultaneously solve multiple goals in multi-goal environments (Andreas et al., 2017; Andrychowicz et al., 2017). In mobile health applications, RL is employed to personalize sequences of treatments, treating each patient as a distinct task (Yom-Tov et al., 2017; Forman et al., 2019; Liao et al., 2020; Ghosh et al., 2023). Many algorithms (Andreas et al., 2017; Andrychowicz et al., 2017; Hessel et al., 2019; Yang et al., 2020) have been designed to jointly learn from multiple tasks. These show significant improvement over those that learn each task individually. To provide explanations for such improvement, recent advancements in Multitask Reinforcement Learning (MTRL) theory study the improved statistical efficiency in estimating unknown parameters by assuming a shared structure across tasks (Brunskill & Li, 2013; Calandriello et al., 2014; Uehara et al., 2021; Xu et al., 2021; Zhang & Wang, 2021; Lu et al., 2021; Agarwal et al., 2022; Cheng et al., 2022; Yang et al., 2022a). Similar setups originate from Multitask Supervised Learning, where it has been shown that learning from multiple tasks reduces the generalization error by a factor of $1/\sqrt{N}$ compared to single-task learning with $N$ being the total number of tasks (Maurer et al., 2016; Du et al., 2020). Nevertheless, these studies overlook an essential aspect of RL, namely exploration.

To understand how learning from multiple tasks, as opposed to single-task learning, could potentially benefit exploration design, we consider a generic MTRL scenario, where an algorithm interacts with a task set $\mathcal{M}$ in rounds $T$. *In each round, the algorithm chooses an exploratory policy $\pi$ that is used to collect one episode worth of data in a task $M \in \mathcal{M}$ of its own choice. A sample-efficient algorithm should output a near-optimal policy for each task in a polynomial number of rounds.*

Exploration design plays an important role in achieving sample-efficient learning. Previous sample-efficient algorithms for single-task learning ($|\mathcal{M}| = 1$) heavily rely on strategic design on the exploratory policies, such as Optimism in Face of Uncertainty (OFU) (Auer et al., 2008; Bartlett & Tewari, 2009; Dann et al., 2017) and Posterior Sampling (Russo & Van Roy, 2014; Osband & Van Roy, 2017). Strategic design is criticized for either being restricted to environments with strong structural assumptions, or involving intractable computation oracle, such as non-convex optimization (Jiang, 2018; Jin et al., 2021a). For instance, GOLF (Jin et al., 2021a), a model-free general function

approximation online learning algorithm, involves finding the optimal Q-value function $f$ in a function class $\mathcal{F}$. It requires the following two intractable computation oracles in (1).

$$\mathcal{F}^t = \{f \in \mathcal{F} : f \text{ has low empirical Bellman error}\} \text{ and } f^{(t)} = \arg\max_{f \in \mathcal{F}^t} f(s_1, \pi(s_1 \mid f)). \quad (1)$$

In contrast, myopic exploration design like $\epsilon$-greedy that injects random noise to a current greedy policy is easy to implement and performs well in a wide range of applications (Mnih et al., 2015; Kalashnikov et al., 2018), while it is shown to have exponential sample complexity in the worst case for single-task learning (Osband et al., 2019). Throughout the paper, we ask the main question:

*Can algorithms with myopic exploration design be sample-efficient for MTRL?*

In this paper, we address the question by showing that *a simple algorithm that explores one task with $\epsilon$-greedy policies from other tasks can be sample-efficient if the task set $\mathcal{M}$ is adequately diverse.* Our results may shed some light on the longstanding mystery that $\epsilon$-greedy is successful in practice, while being shown sample-inefficient in theory. We argue that in an MTRL setting, $\epsilon$-greedy policies may no longer behave myopically, as they explore myopically around the optimal policies from other tasks. When the task set is adequately diverse, this exploration may provide sufficient coverage. It is worth-noting that this may partially explain the success of curriculum learning in RL (Narvekar et al., 2020), that is the optimal policy of one task may easily produce a good exploration for the next task, thus reducing the complexity of exploration. This connection will be formally discussed later.

To summarize our contributions, *this work provides a general framework that converts the task of strategic exploration design into the task of constructing diverse task set in an MTRL setting*, where myopic exploration like $\epsilon$-greedy can also be sample-efficient. We discuss a sufficient diversity condition under a general value function approximation setting (Jin et al., 2021a; Dann et al., 2022). We show that the condition guarantees polynomial sample-complexity bound by running the aforementioned algorithm with myopic exploration design. We further discuss how to satisfy the diversity condition in different case studies, including tabular cases, linear cases and linear quadratic regulator cases. In the end, we validate our theory with experiments on synthetic robotic control environments, where we observe that a diverse task set is similar to the task selection of the state-of-the-art automatic curriculum learning algorithm, which has been empirically shown to improve sample efficiency.

## 2 PROBLEM SETUP

The following are notations that will be used throughout the paper.

**Notation.** For a positive integer $H$, we denote $[H] := \{1, \ldots, H\}$. For a discrete set $\mathcal{A}$, we denote $\Delta_{\mathcal{A}}$ by the set of distributions over $\mathcal{A}$. We use $\mathcal{O}$ and $\Omega$ to denote the asymptotic upper and lower bound notations and use $\tilde{\mathcal{O}}$ and $\tilde{\Omega}$ to hide the logarithmic dependence. Let $\{\mathbf{1}_i\}_{i \in [d]}$ be the standard basis that spans $\mathbb{R}^d$. We let $N_{\mathcal{F}}(\rho)$ denote the $\ell_\infty$ covering number of a function class $\mathcal{F}$ at scale $\rho$. For a class $\mathcal{F}$, we denote the $N$-times Cartesian product of $\mathcal{F}$ by $(\mathcal{F})^{\otimes N}$.

### 2.1 PROPOSED MULTITASK LEARNING SCENARIO

Throughout the paper, we consider each task as an episodic MDP denoted by $M = (\mathcal{S}, \mathcal{A}, H, P_M, R_M)$, where $\mathcal{S}$ is the state space, $\mathcal{A}$ is the action space, $H \in \mathbb{N}$ represents the horizon length in each episode, $P_M = (P_{h,M})_{h \in [H]}$ is the collection of transition kernels, and $R_M = (R_{h,M})_{h \in [H]}$ is the collection of immediate reward functions. Each $P_{h,M} : \mathcal{S} \times \mathcal{A} \mapsto \Delta(\mathcal{S})$ and each $R_{h,M} : \mathcal{S} \times \mathcal{A} \mapsto [0, 1]$. Note that we consider a scenario in which all the tasks share the same state space, action space, and horizon length.

An agent interacts with an MDP $M$ in the following way: starting with a fixed initial state $s_1$, at each step $h \in [H]$, the agent decides an action $a_h$ and the environment samples the next state $s_{h+1} \sim P_{h,M}(\cdot \mid s_h, a_h)$ and next reward $r_h = R_{h,M}(s_h, a_h)$. An episode is a sequence of states, actions, and rewards $(s_1, a_1, r_1, \ldots, s_H, a_H, r_H, s_{H+1})$. In general, we assume that the sum of $r_h$ is upper bounded by 1 for any action sequence almost surely. The goal of an agent is to maximize the cumulative reward $\sum_{h=1}^{H} r_h$ by optimizing their actions.

The agent chooses actions based on *Markovian policies* denoted by $\pi = (\pi_h)_{h \in [H]}$ and each $\pi_h$ is a mapping $\mathcal{S} \mapsto \Delta_{\mathcal{A}}$, where $\Delta_{\mathcal{A}}$ is the set of all distributions over $\mathcal{A}$. Let $\Pi$ denote the space of all such policies. For a finite action space, we let $\pi_h(a \mid s)$ denote the probability of selecting action $a$ given state $s$ at the step $h$. In case of the infinite action space, we slightly abuse the notation by letting $\pi_h(\cdot \mid s)$ denote the density function.

**Proposed multitask learning scenario and objective.** We consider the following multitask RL learning scenario. An algorithm interacts with a set of tasks $\mathcal{M}$ sequentially for $T$ rounds. At the each round $t$, the algorithm chooses an exploratory policy, which is used to collect data for one episode in a task $M \in \mathcal{M}$ of its own choice. At the end of $T$ rounds, the algorithm outputs a set of policies $\{\pi_M\}_{M \in \mathcal{M}}$. The goal of an algorithm is to learn a near-optimal policy $\pi_M$ for each task $M \in \mathcal{M}$. The sample complexity of an algorithm is defined as follows.

**Definition 1** (MTRL Sample Complexity). *An algorithm $\mathcal{L}$ is said to have sample-complexity of $\mathcal{C}_{\mathcal{M}}^{\mathcal{L}} : \mathbb{R} \times \mathbb{R} \mapsto \mathbb{N}$ for a task set $\mathcal{M}$ if for any $\beta > 0, \delta \in (0, 1)$, it outputs a $\beta$-optimal policy $\pi_M$ for each MDP $M \in \mathcal{M}$ with probability at least $1 - \delta$, by interacting with the task set for $\mathcal{C}_{\mathcal{M}}^{\mathcal{L}}(\beta, \delta)$ rounds. We omit the notations for $\mathcal{L}$ and $\mathcal{M}$ when they are clear from the context to ease presentation.*

A sample-efficient algorithm should have a sample-complexity polynomial in the parameters of interests. For the tabular case, where the state space and action space are finite, $\mathcal{C}(\beta, \delta)$ should be polynomial in $|\mathcal{S}|, |\mathcal{A}|, |\mathcal{M}|$, H, and $1/\beta$ for a sample-efficient learning. Current state-of-the-art algorithm (Zhang et al., 2021) on a single-task tabular MDP achieves sample-complexity of $\tilde{\mathcal{O}}(|\mathcal{S}||\mathcal{A}|/\beta^2)$[1]. This bound translates to a MTRL sample-complexity bound of $\tilde{\mathcal{O}}(|\mathcal{M}||\mathcal{S}||\mathcal{A}|/\beta^2)$ by running their algorithm individually for each $M \in \mathcal{M}$. However, their exploration design closely follows the principle of Optimism in Face of Uncertainty, which is normally criticized for over-exploring.

## 2.2 VALUE FUNCTION APPROXIMATION

We consider the setting where value functions are approximated by general function classes. Denote the value function of an MDP $M$ with respect to a policy $\pi$ by

$$Q_{h,M}^{\pi}(s, a) = \mathbb{E}_{\pi}^{M} \left[ r_h + V_{h+1,M}^{\pi}(s_{h+1}) \mid s_h = s, a_h = a \right]$$
$$V_{h,M}^{\pi}(s) = \mathbb{E}_{\pi}^{M} \left[ Q_{h,M}^{\pi}(s_h, a_h) \mid s_h = s \right],$$

where by $\mathbb{E}_{\pi}^{M}$, we take expectation over the randomness of trajectories sampled by policy $\pi$ on MDP $M$ and we let $V_{H+1,M}^{\pi}(s) \equiv 0$ for all $s \in \mathcal{S}$ and $\pi \in \Pi$. We denote the optimal policy for MDP $M$ by $\pi_M^*$. The corresponding value functions are denoted by $V_{h,M}^*$ and $Q_{h,M}^*$, which is shown to satisfy Bellman Equation $\mathcal{T}_h^M Q_{h+1,M}^* = Q_{h,M}^*$, where for any $g : \mathcal{S} \times \mathcal{A} \mapsto [0, 1]$, $\left( \mathcal{T}_h^M g \right)(s, a) = \mathbb{E}[r_h + \max_{a' \in \mathcal{A}} g(s_{h+1}, a') \mid s_h = s, a_h = a]$.

The agent has access to a collection of function classes $\mathcal{F} = (\mathcal{F}_h : \mathcal{S} \times \mathcal{A} \mapsto [0, H])_{h \in [H+1]}$. We assume that different tasks share the same set of function class. For each $f \in \mathcal{F}$, we denote by $f_h \in \mathcal{F}_h$ the $h$-th component of the function $f$. We let $\pi^f = \{\pi_h^f\}_{h \in [H]}$ be the greedy policy with $\pi_h^f(s) = \arg\max_{a \in \mathcal{A}} f_h(s, a)$. When it is clear from the context, we slightly abuse the notation and let $f \in (\mathcal{F})^{\otimes |\mathcal{M}|}$ be a joint function for all the tasks. We further let $f_M$ denote the function for the task $M$ and $f_{h,M}$ denote its $h$-th component.

Define Bellman error operator $\mathcal{E}_h^M$ such that $\mathcal{E}_h^M f = f_h - \mathcal{T}_h^M f_{h+1}$ for any $f \in \mathcal{F}$. The goal of the learning algorithm is to approximate $Q_{h,M}^*$ through the function class $\mathcal{F}_h$ by minimizing the empirical Bellman error for each step $h$ and task $M$.

To provide theoretical guarantee on this practice, we make the following realizability and completeness assumptions. The two assumptions and their variants are commonly used in the literature (Dann et al., 2017; Jin et al., 2021a).

**Assumption 1** (Realizability and Completeness). *For any MDP $M$ considered in this paper, we assume $\mathcal{F}$ is realizable and complete under the Bellman operator such that $Q_{h,M}^* \in \mathcal{F}_h$ for all $h \in [H]$ and for every $h \in [H]$, $f_{h+1} \in \mathcal{F}_{h+1}$ there is a $f_h \in \mathcal{F}_h$ such that $f_h = \mathcal{T}_h^M f_{h+1}$.*

---

[1]This bound is under the regime with $1/\beta \gg |\mathcal{S}|$

## 2.3 Myopic Exploration Design

As opposed to carefully designed exploration, myopic exploration injects random noise to the current greedy policy. For a given greedy policy $\pi$, we use $\mathrm{expl}(\pi)$ to denote the myopic exploration policy based on $\pi$. Depending on the action space, the function $\mathrm{expl}$ can take different forms. The most common choice for finite action spaces is $\epsilon$-greedy, which mixes the greedy policy with a random action: $\mathrm{expl}(\pi_h)(a \mid s) = (1-\epsilon_h)\pi_h(a \mid s) + \epsilon_h/A$.[2] As it is our main study of exploration strategies, we let $\mathrm{expl}$ be $\epsilon$-greedy function if not explicitly specified. For a continuous action space, we consider exploration with Gaussian noise: $\mathrm{expl}(\pi_h)(a \mid s) = (1-\epsilon_h)\pi_h(a \mid s) + \epsilon_h \exp(-a^2/2\sigma_h^2)/\sqrt{2\pi\sigma_h^2}$. Gaussian noise is useful for Linear Quadratic Regulator (LQR) setting (discussed in Appendix F.)

## 3 Multitask RL Algorithm with Policy-Sharing

In this section, we introduce a generic algorithm (Algorithm 1) for the proposed multitask RL scenario without any strategic exploration, whose theoretical properties will be studied throughout the paper. In a typical single-task learning, a myopically exploring agent samples trajectories by running its current greedy policy estimated from the historical data equipped with naive explorations like $\epsilon$-greedy.

In light of the exploration benefits of MTRL, we study Algorithm 1 as a counterpart of the single-task learning scenario in the MTRL setting. Algorithm 1 maintains a dataset for each MDP separately and different tasks interact in the following way: in each round, Algorithm 1 explores every MDP with an exploratory policy that is the mixture (defined in Definition 2) of greedy policies of all the MDPs in the task set (Line 8). One way to interpret Algorithm 1 is that we share knowledge across tasks by policy sharing instead of parameter sharing or feature extractor sharing in the previous literature.

**Definition 2** (Mixture Policy). *For a set of policies $\{\pi_i\}_{i=1}^N$, we denote $\mathrm{Mixture}(\{\pi_i\}_{i=1}^N)$ by the mixture of $N$ policies, such that before the start of an episode, it samples $I \sim \mathrm{Unif}([N])$, then runs policy $\pi_I$ for the rest of the episode.*

The greedy policy is obtained from an offline learning oracle $\mathcal{Q}$ (Line 4) that maps a dataset $\mathcal{D} = \{(s_i, a_i, r_i, s_i')\}_{i=1}^N$ to a function $f \in \mathcal{F}$, such that $\mathcal{Q}(\mathcal{D})$ is an approximate solution to the following minimization problem $\arg\min_{f \in \mathcal{F}} \sum_{i=1}^N (f_{h_i}(s_i, a_i) - r_i - \max_{a' \in \mathcal{A}} f_{h_i+1}(s_i', a'))^2$. In practice, one can run fitted Q-iteration for an approximate solution.

**Connection to Hindsight Experience Replay (HER) and multi-goal RL.** We provide some justifications for the choice of the mixture policy. HER is a common practice (Andrychowicz et al., 2017) in the multi-goal RL setting (Andrychowicz et al., 2017; Chane-Sane et al., 2021; Liu et al., 2022), where the reward distribution is a function of goal-parameters. HER relabels the rewards in trajectories in the experience buffer such that they were as if sampled for a different task. Notably, this exploration strategy is akin to randomly selecting task and collecting a trajectory on using its own epsilon-greedy policy, followed by relabeling rewards to simulate a trajectory on, which is equivalent to Algorithm 1. Yang et al. (2022b); Zhumabekov et al. (2023) also designed an algorithm with ensemble policies, which is shown to improve generalizability. It is worth noting that the ensemble implementation of Algorithm 1 and that of Zhumabekov et al. (2023) differs. In Algorithm 1, policies are mixed trajectory-wise, with one policy randomly selected for the entire episode. In contrast, Zhumabekov et al. (2023) mixes policies on a step-wise basis for a continuous action space, selecting actions as a weighted average of actions chosen by different policies. This distinction is vital in showing the rigorous theoretical guarantee outlined in this paper.

**Connection to curriculum learning.** Curriculum learning is the approach that learns tasks in a specific order to improve the multitask learning performance (Bengio et al., 2009). Although Algorithm 1 does not explicitly implement curriculum learning by assigning preferences to tasks, improvement could be achieved through adaptive task selection that may reflect the benefits of curriculum learning. Intuitively, any curricula that selects tasks through an order of $M_1, \ldots, M_T$ is implicitly included in Algorithm 1 as it explores all the MDPs in each round with the mixture of all epsilon-greedy policies. This means that the sample-complexity of Algorithm 1 provides an upper bound on the sample complexity of underlying optimal task selection. A formal discussion is deferred to Appendix B.

---

[2]Note that we consider a more general setup, where the exploration probability $\epsilon$ can depend on $h$.

---

**Algorithm 1** Generic Algorithm for MTRL with Policy-Sharing

---

1: **Input:** function class $\mathcal{F} = \mathcal{F}_1 \times \cdots \times \mathcal{F}_{H+1}$, task set $\mathcal{M}$, exploration function $\mathrm{expl}$
2: Initialize $\mathcal{D}_{0,M} \leftarrow \emptyset$ for all $M \in \mathcal{M}$
3: **for** round $t = 1, 2, \ldots, \lfloor T/|\mathcal{M}| \rfloor$ **do**
4:      Offline learning oracle outputs $\hat{f}_{t,M} \leftarrow \mathcal{Q}(\mathcal{D}_{t-1,M})$ for each $M$        $\triangleright$ Offline learning
5:      Set myopic exploration policy $\hat{\pi}_{t,M} \leftarrow \mathrm{expl}(\pi^{\hat{f}_{t,M}})$ for each $M$
6:      Set $\hat{\pi}_t \leftarrow \mathrm{Mixture}(\{\hat{\pi}_{t,M}\}_{M \in \mathcal{M}})$        $\triangleright$ Share policies
7:      **for** $M \in \mathcal{M}$ **do**
8:          Sample one episode $\tau_{t,M}$ on MDP $M$ with policy $\hat{\pi}_t$        $\triangleright$ Collect new trajectory
9:          Add $\tau_{t,M}$ to the dataset: $\mathcal{D}_{t,M} \leftarrow \mathcal{D}_{t-1,M} \cup \{\tau_{t,M}\}$
10:      **end for**
11: **end for**
12: **Return** $\hat{\pi}_M = \mathrm{Mixture}(\{\hat{\pi}_{t,M}\}_{t \in \lfloor T/|\mathcal{M}| \rfloor})$ for each $M$

---

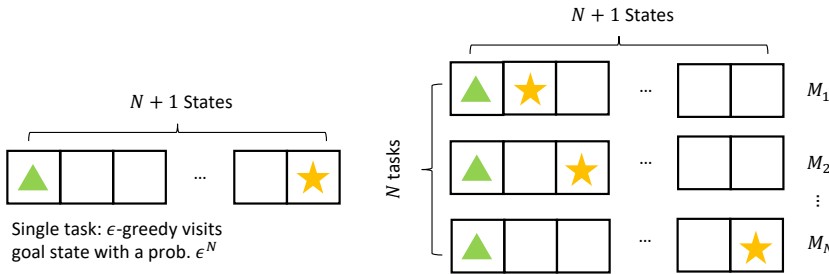

Figure 1: A diverse grid-world task set on a long hallway with $N + 1$ states. From the left to the right, it represents a single-task and a multitask learning scenario, respectively. The triangles represent the starting state and the stars represent the goal states, where an agent receives a positive reward. The agent can choose to move forward or backward.

## 4    GENERIC SAMPLE COMPLEXITY GUARANTEE

In this section, we rigorously define the diversity condition and provide a sample-complexity upper bound for Algorithm 1. We start with introducing an intuitive example on how diversity encourages exploration in a multitask setting.

**Motivating example.** Figure 1 introduces a motivating example of grid-world environment on a long hallway with $N + 1$ states. Since this is a deterministic tabular environment, whenever a task collects an episode that visits its goal state, running an offline policy optimization algorithm with pessimism will output its optimal policy.

Left penal of Figure 1 is a single-task learning, where the goal state is $N$ steps away from the initial state, making it exponentially hard to visit the goal state with $\epsilon$-greedy exploration. Figure 1 on the right demonstrates a multitask learning scenario with $N$ tasks, whose goal states "diversely" distribute along the hallway. A main message of this paper is the advantage of exploring one task by running the $\epsilon$-greedy policies of other tasks. To see this, consider any current greedy policies $(\pi_1, \pi_2, \ldots, \pi_N)$. Let $i$ be the first non-optimal policy, i.e. all $j < i$, $\pi_j$ is optimal for $M_j$. Since $\pi_{i-1}$ is optimal, by running an $\epsilon$-greedy of $\pi_{i-1}$ on MDP $M_i$, we have a probability of $\prod_{h=1}^{i-1}(1 - \epsilon_h)\epsilon_i$ to visit the goal state of $M_i$, allowing it to improve its policy to optimal in the next round. Such improvement can only happen for $N$ times and all policies will be optimal within polynomial (in $N$) number of rounds if we choose $\epsilon_h = 1/(h + 1)$. Hence, myopic exploration with a diverse task set leads to sample-efficient learning. The rest of the section can be seen as generalizing this idea to function approximation.

### 4.1    MULTITASK MYOPIC EXPLORATION GAP

Dann et al. (2022) proposed an assumption named Myopic Exploration Gap (MEG) that allows efficient myopic exploration for a single MDP under strong assumptions on the reward function, or on the mixing time. We extend this definition to the multitask learning setting. For the conciseness of the notation, we let $\mathrm{expl}(f)$ denote the following mixture policy $\mathrm{Mixture}(\{\mathrm{expl}(\pi^{f_M})\}_{M \in \mathcal{M}})$

for a joint function $f \in (\mathcal{F})^{\otimes|\mathcal{M}|}$. Intuitively, a large myopic exploration gap implies that within all the policies that can be learned by the current exploratory policy, there exists one that can make significant improvement on the current greedy policy.

**Definition 3** (Multitask Myopic Exploration Gap (Multitask MEG)). *For any $\mathcal{M}$, a function class $\mathcal{F}$, a joint function $f \in (\mathcal{F})^{\otimes|\mathcal{M}|}$, we say that $f$ has $\alpha(f, \mathcal{M}, \mathcal{F})$-myopic exploration gap, where $\alpha(f, \mathcal{M}, \mathcal{F})$ is the value to the following maximization problem:*

$$\max_{M \in \mathcal{M}} \sup_{\tilde{f} \in \mathcal{F}, c \geq 1} \frac{1}{\sqrt{c}}(V_{1,M}^{\tilde{f}} - V_{1,M}^{f_M}), \text{ s.t. for all } f' \in \mathcal{F} \text{ and } h \in [H],$$

$$\mathbb{E}_{\pi^{\tilde{f}}}^M[(\mathcal{E}_h^M f')(s_h, a_h)]^2 \leq c\mathbb{E}_{\text{expl}(f)}^M[(\mathcal{E}_h^M f')(s_h, a_h)]^2$$

$$\mathbb{E}_{\pi^{f_M}}^M[(\mathcal{E}_h^M f')(s_h, a_h)]^2 \leq c\mathbb{E}_{\text{expl}(f)}^M[(\mathcal{E}_h^M f')(s_h, a_h)]^2.$$

*Let $M(f, \mathcal{M}, \mathcal{F})$, $c(f, \mathcal{M}, \mathcal{F})$ be the corresponding $M \in \mathcal{M}$ and $c$ that attains the maximization.*

**Design of myopic exploration gap.** To illustrate the spirit of this definition, we specialize to the tabular case, where conditions in Definition 3 can be replaced by the concentrability (Jin et al., 2021b) condition: for all $s, a, h \in \mathcal{S} \times \mathcal{A} \times [H]$, we require

$$\mu_{h,M}^{\pi^{\tilde{f}}}(s, a) \leq c\mu_{h,M}^{\text{expl}(f)}(s, a) \text{ and } \mu_{h,M}^{\pi^{f_M}}(s, a) \leq c\mu_{h,M}^{\text{expl}(f)}(s, a), \tag{2}$$

where $\mu_{h,M}^{\pi}(s, a)$ is the occupancy measure, i.e. the probability of visiting $(s, a)$ at the step $h$ by running policy $\pi$ on MDP $M$. The design of myopic exploration gap connects deeply to the theory of offline Reinforcement Learning. For a specific MDP $M$, Equation (2) defines a set of policies with concentrability assumption (Xie et al., 2021) that can be accurately evaluated through the offline data collected by the current behavior policy. As an extension to the Single-task MEG in Dann et al. (2022), Multitask MEG considers the maximum myopic exploration gap over a set of MDPs and the behavior policy is a mixture of all the greedy policies. Definition 3 reduces to the Single-task MEG when the task set $\mathcal{M}$ is a singleton.

### 4.2 SAMPLE COMPLEXITY GUARANTEE

We propose Diversity in Definition 5, which relies on having a lower bounded Multitask MEG for any suboptimal policy. We then present Theorem 1 that shows an upper bound for sample complexity of Algorithm 1 by assuming diversity.

**Definition 4** (Multitask Suboptimality). *For a multitask RL problem with MDP set $\mathcal{M}$ and value function class $\mathcal{F}$. Let $\mathcal{F}_\beta \subset (\mathcal{F})^{\otimes|\mathcal{M}|}$ be the $\beta$-suboptimal class, such that for any $f \in \mathcal{F}_\beta$, there exists $f_M$ and $\pi^{f_M}$ is $\beta$-suboptimal for MDP $M$, i.e. $V_{1,M}^{\pi^{f_M}} \leq \max_{\pi \in \Pi} V_{1,M}^{\pi} - \beta$.*

**Definition 5** (Diverse Tasks). *For some function $\tilde{\alpha} : [0, 1] \mapsto \mathbb{R}$, and $\tilde{c} : [0, 1] \mapsto \mathbb{R}$, we say that a tasks set is $(\tilde{\alpha}, \tilde{c})$-diverse if any $f \in \mathcal{F}_\beta$ has multitask myopic exploration gap $\alpha(f, \mathcal{M}, \mathcal{F}) \geq \tilde{\alpha}(\beta)$ and $c(f, \mathcal{M}, \mathcal{F}) \leq \tilde{c}(\beta)$ for any constant $\beta > 0$.*

To simplify presentation, we state the result here assuming parametric growth of the Bellman eluder dimension and covering number, that is $d_{\text{BE}}(\mathcal{F}, \Pi_\mathcal{F}, \rho) \leq d_\mathcal{F}^{\text{BE}} \log(1/\rho)$ and $\log(N'(\mathcal{F})(\rho)) \leq d_\mathcal{F}^{\text{cover}} \log(1/\rho)$, where $\dim_{\text{BE}}(\mathcal{F}, \Pi_\mathcal{F}, \rho)$ is the Bellman-Eluder dimension of class $\mathcal{F}$ and $N'_\mathcal{F}(\rho) = \sum_{h=1}^{H-1} N_{\mathcal{F}_h}(\rho)N_{\mathcal{F}_{h+1}}(\rho)$. A formal definition of Bellman-Eluder dimension is deferred to Appendix H. This parametric growth rate holds for most of the regular classes including tabular and linear (Russo & Van Roy, 2013). Similar assumptions are made in Chen et al. (2022a).

**Theorem 1** (Upper Bound for Sample Complexity). *Consider a multitask RL problem with MDP set $\mathcal{M}$ and value function class $\mathcal{F}$ such that $\mathcal{M}$ is $(\tilde{\alpha}, \tilde{c})$-diverse. Then Algorithm 1 with $\epsilon$-greedy exploration function has a sample-complexity*

$$\mathcal{C}(\beta, \delta) = \tilde{\mathcal{O}}\left(|\mathcal{M}|^2 H^2 d_\mathcal{F}^{BE} d_\mathcal{F}^{cover} \frac{\ln \tilde{c}(\beta)}{\tilde{\alpha}^2(\beta)} \ln(1/\delta)\right).$$

### 4.3 COMPARING SINGLE-TASK AND MULTITASK MEG

The sample complexity bound in Theorem 1 reduces to the single-task sample complexity in Dann et al. (2022) when $\mathcal{M}$ is a singleton. To showcase the potential benefits of multitask learning, we provide

a comprehensive comparison between Single-task and Multitask MEG. We focus our discussion on $\alpha(f, \mathcal{M}, \mathcal{F})$ because $c(f, \mathcal{M}, \mathcal{F}) \leq ((\max_\pi V_{1,M(f,\mathcal{M},\mathcal{F})}^\pi - V_{1,M(f,\mathcal{M},\mathcal{F})}^{\pi^{f_M}})/\alpha(f, \mathcal{M}, \mathcal{F}))^2 \leq 1/\alpha^2(f, \mathcal{M}, \mathcal{F})$ and $c(f, \mathcal{M}, \mathcal{F})$ only impacts sample complexity bound through a logarithmic term.

We first show that Multitask MEG is lower bounded by Single-task MEG up to a factor of $1/\sqrt{|\mathcal{M}|}$.

**Proposition 1.** *Let $\mathcal{M}$ be any set of MDPs and $\mathcal{F}$ be any function class. We have that $\alpha(f, \mathcal{M}, \mathcal{F}) \geq \alpha(f_M, \{M\}, \mathcal{F})/\sqrt{|\mathcal{M}|}$ for all $f \in (\mathcal{F})^{\otimes|\mathcal{M}|}$ and $M \in \mathcal{M}$.*

Proposition 1 immediately implies that whenever all tasks in a task set can be learned with myopic exploration individually (see examples in Dann et al. (2022)), they can also be learned with myopic exploration through MTRL with Algorithm 1 with an extra factor of $|\mathcal{M}|^2$, which is still polynomial in all the related parameters.

We further argue that Single-task MEG can easily be exponentially small in $H$, in which case, myopic exploration fails.

**Proposition 2.** *Let $M$ be a tabular sparse-reward MDP, i.e., $R_{h,M}(s,a) = 0$ at all $(s,a,h)$ except for a goal tuple $(s_t, a_t, h_t)$ and $R_{h_t,M}(s_t, a_t) = 1$. Recall $\mu_{h,M}^\pi(s,a) = \mathbb{E}_\pi^M[\mathbb{1}_{s_h=s,a_h=a}]$. Then*

$$\alpha(f, \{M\}, \mathcal{F}) \leq \max_{\tilde{\pi}}(\mu_{h_t}^{\tilde{\pi}}(s_t, a_t) - \mu_{h_t}^{\pi^f}(s_t, a_t))\sqrt{\mu_{h_t}^{\text{expl}(\pi^f)}(s_t, a_t)/\mu_{h_t}^{\tilde{\pi}}(s_t, a_t)} \leq \sqrt{\mu_{h_t}^{\text{expl}(\pi^f)}(s_t, a_t)}.$$

Sparse-reward MDP is widely studied in goal-conditioned RL, where the agent receives a non-zero reward only when it reaches a goal-state (Andrychowicz et al., 2017; Chane-Sane et al., 2021). Proposition 2 implies that a single-task MEG can easily be exponentially small in $H$ as long as one can find a policy $\pi$ such that its $\epsilon$-greedy version, $\text{expl}(\pi)$, visits the goal tuple $(s_t, a_t, h_t)$ with a probability that is exponentially small in $H$. This is true when the environment requires the agent to execute a fixed sequence of actions to reach the goal tuple as it is the case in Figure 1. Indeed, in the environment described in the left panel of Figure 1, a policy that always moves left will have $\alpha(f, \{M\}, \mathcal{F}) \leq \sqrt{\mu_{h_t}^{\text{expl}(\pi^f)}(s_t, a_t)} \leq \sqrt{\Pi_{h=1}^H(\epsilon_h/2)} \leq 2^{-H/2}$. This is also consistent with our previous discussion that $\epsilon$-greedy requires $\Omega(2^H)$ number of episodes to learn the optimal policy in the worst case. As we will show later in Section 5.1, Multitask MEG for the tabular case can be lowered bounded by $\Omega(\sqrt{1/(|\mathcal{A}||\mathcal{M}|H)})$ for adequately diverse task set $\mathcal{M}$, leading to an exponential separation between Single-task and Multitask MEG.

## 5 LOWER BOUNDING MYOPIC EXPLORATION GAP

Following the generic result in Theorem 1, the key to the problem is to lower bound myopic exploration gap $\tilde{\alpha}(\beta)$. In this section, we lower bound $\tilde{\alpha}(\beta)$ for the linear MDP case. We defer an improved analysis for the tabular case and the Linear Quadratic Regulator cases to Appendix F.

Linear MDPs have been an important case study for the theory of RL (Wang et al., 2019; Jin et al., 2020b; Chen et al., 2022b). It is a more general case than tabular MDP and has strong implication for Deep RL. In order to employ $\epsilon$-greedy, we consider finite action space, while the state space can be infinite.

**Definition 6** (Linear MDP Jin et al. (2020b)). *An MDP is called linear MDP if its transition probability and reward function admit the following form. $P_h(s' \mid s, a) = \langle \phi_h(s,a), \mu_h(s') \rangle$ for some known function $\phi_h : \mathcal{S} \times \mathcal{A} \mapsto (\mathbb{R}^+)^d$ and unknown function $\mu_h : \mathcal{S} \mapsto (\mathbb{R}^+)^d$. $R_h(s,a) = \langle \phi_h(s,a), \theta_h \rangle$ for unknown parameters $\theta_h$ [3]. Without loss of generality, we assume $\|\phi_h(s,a)\| \leq 1$ for all $s, a, h \in \mathcal{S} \times \mathcal{A} \times \mathcal{H}$ and $\max\{\|\mu_h(s)\|, \|\theta_h\|\} \leq \sqrt{d}$ for all $s, h \in \mathcal{S} \times [H]$.*

An important property of Linear MDPs is that the value function also takes the linear form and the linear function class defined below satisfies Assumption 1.

**Proposition 3** (Proposition 2.3 (Jin et al., 2020b)). *For linear MDPs, we have for any policy $\pi$, $Q_{h,M}^\pi(s,a) = \langle \phi_h(s,a), w_{h,M}^\pi \rangle$, where $w_{h,M}^\pi = \theta_{h,M} + \int_\mathcal{S} V_{h+1,M}^\pi(s')\mu_h(s')ds' \in \mathbb{R}^d$. Therefore, we only need to consider $\mathcal{F}_h = \{(s,a) \mapsto \langle \phi_h(s,a), w \rangle : w \in \mathbb{R}^d, \|w\|_2 \leq 2\sqrt{d}\}$.*

Now we are ready to define a diverse set of MDPs for the linear MDP case.

---

[3]Note that we consider non-negative measure $\mu_h$.

**Definition 7** (Diverse MDPs for linear MDP case). *We say $\mathcal{M}$ is a diverse set of MDPs for the linear MDP case, if they share the same feature extractor $\phi_h$ and the same measure $\mu_h$ (leading to the same transition probabilities) and for any $h \in [H]$, there exists a subset $\{M_{i,h}\}_{i\in[d]} \subset \mathcal{M}$, such that the reward parameter $\theta_{h,M_{i,h}} = \mathbf{1}_i$ and all the other $\theta_{h',M_{i,h}} = 0$ with $h' \neq h$, where $\mathbf{1}_i$ is the onehot vector with a positive entry at the dimension $i$.*

We need the assumption that the minimum eigenvalue of the covariance matrix is strictly lower bounded away from 0. The feature coverage assumption is commonly use in the literature that studies Linear MDPs (Agarwal et al., 2022). Suppose Assumption 2 hold, we have Theorem 2, which lower bounds the multitask myopic exploration gap. Combined with Theorem 1, we have a sample-complexity bound of $\tilde{\mathcal{O}}(|\mathcal{M}|^3 H^3 d^2 |\mathcal{A}|/(\beta^2 b_1^2))$ with $|\mathcal{M}| \geq d$.

**Assumption 2** (Feature coverage). *For any $\nu \in \mathbb{S}^{d-1}$ and $[\nu]_i > 0$ for all $i \in [d]$, there exists a policy $\pi$ such that $\mathbb{E}_\pi[\nu^\top \phi_h(s_h, a_h)] \geq b_1$, for some constant $b_1 > 0$.*

**Theorem 2.** *Consider $\mathcal{M}$ to be a diverse set as in Definition 7. Suppose Assumption 2 holds and $\beta \leq b_1/2$, then we have for any $f \in \mathcal{F}_\beta$, $\alpha(f, \mathcal{F}, \mathcal{M}) = \Omega(\sqrt{\beta^2 b_1^2/(|\mathcal{A}||\mathcal{M}|H})$ by setting $\epsilon_h = 1/(h+1)$.*

Proof of Theorem 2 critically relies on iteratively showing the following lemma, which can be interpreted as showing that the feature covariance matrix induced by the optimal policies is full rank.

**Lemma 1.** *Fix a step $h$ and fix a $\beta < b_1/2$. Let $\{\pi_i\}_{i=1}^d$ be $d$ policies such that $\pi_i$ is a $\beta$-optimal policy for $M_{i,h}$ as in Definition 7. Let $\tilde{\pi} = \mathrm{Mixture}(\{\mathrm{expl}(\pi_i)\}_{i=1}^d)$. Then we have $\lambda_{\min}(\Phi_{h+1}^{\tilde{\pi}}) \geq \epsilon_h \prod_{h'=1}^{h-1}(1 - \epsilon_{h'}) b_1^2/(2dA)$.*

### 5.1 DISCUSSIONS ON THE TABULAR CASE

Diverse tasks in Definition 7, when specialized to the tabular case, corresponds to $S \times H$ sparse-reward MDPs. Interestingly, similar constructions are used in reward-free exploration (Jin et al., 2020a), which shows that by calling an online-learning oracle individually for all the sparse reward MDPs, one can generate a dataset that outputs a near-optimal policy for any given reward function. We want to point out the intrinsic connection between the two settings: our algorithm, instead of generating an offline dataset all at once, generates an offline dataset at each step $h$ that is sufficient to learn a near-optimal policy for MDPs that corresponds to the step $h + 1$.

**Relaxing coverage assumption.** Though feature coverage Assumption (Assumption 2) is handy for our proof as it guarantees that any $\beta$-optimal policy (with $\beta < b_1/2$) has a probability at least $b_1/2$ to visit their goal state, this assumption may not be reasonable for the tabular MDP case. Intuitively, without this assumption, a $\beta$-optimal policy can be an arbitrary policy and we can have at most $S$ such policies in total leading to a cumulative error of $S\beta$. A naive solution is to request a $S^{-H}\beta$ accuracy at the first step, which leads to exponential sample-complexity. In Appendix G, we show that an improved analysis can be done for the tabular MDP without having to make the coverage assumption. However, an extra price of $SH$ has to be paid.

## 6 IMPLICATIONS OF DIVERSITY IN DEEP RL

Though this is a paper focusing on the theoretical justification, we add some preliminary discussion on what diversity means in Deep RL. Thus far, we have provided concrete example for diverse task set in tabular and linear MDPs with lower bounded Multitask MEG. It is often the case that in Deep RL, a pretrained feature extractor is used to generate embeddings for Q-value function and then a linear mapping is applied to generate the final output (Bhateja et al., 2023; Hejna III & Sadigh, 2023). This manner is similar to the setup in linear MDPs. We have shown that to achieve diversity for linear MDPs, one important property is to have a full rank covariance matrix of the embeddings at each step $h$ if the optimal policy is executed (Lemma 1). In this section, we conduct simple simulation studies on BipedalWalker (Portelas et al., 2020), a robotic control environment, to verify that whether a more spread spectrum of the covariance matrix of the embeddings would lead to better sample efficiency.

We found in the experiments that *the subset of tasks that leads to more spread spectrum is similar to the task selection of automatic curriculum learning, which has been empirically shown to learn more sample efficiently.* Due to the page limit we provide a high-level summary of the simulation study and the readers can find full details in Appendix I.

**Experiment setup.** The learning agent is embodied into a bipedal walker whose motors are controlled by continuous actions of torques. The objective of the agent is to move forward as far as possible,

while crossing stumps with varying heights at regular intervals. An environment or task, denoted as $M_{p,q}$, is controlled by a parameter vector $(p,q)$, where $p$ and $q$ denote the heights of the stumps and the spacings between the stumps, respectively. Intuitively, challenging environments have higher and denser stumps. The agent is trained by Proximal Policy Optimization (PPO) (Schulman et al., 2017) with a standard actor-critic framework (Konda & Tsitsiklis, 1999) and with Boltzmann exploration that regularizes entropy. Detailed training setup can be found in Appendix I.

**Proper level of heights leads to more spread spectrum.** We evaluate the extracted feature at the end of the training generated by near-optimal policies on 100 tasks with different parameter vectors $(p,q)$. We then compute the covariance matrix of the features for each task, whose spectrum are shown in Figure 2 (a). We observe that the eigenvalues can be 10 times higher for environments with an appropriate height (1.0-2.3), compared to extremely high and low heights, while they are roughly the same at different levels of spacings. This indicates that choosing an appropriate height is the key to properly scheduling tasks.

**Coincidence with automatic curriculum learning.** In fact, the task selection is similar to the tasks selected by the state-of-the-art Automatic Curriculum Learning (ACL). We investigate the curricula generated by ALP-GMM (Portelas et al., 2020), a well-established curriculum learning algorithm, for training an agent in the BipedalWalker environment for 20 million timesteps. Figure 2 (b) gives the density plots of the ACL task sampler during the training process, which shows a significant preference over heights in the middle range, with little preference over spacing. Since ACL has been shown to significantly improve the learning performance in BipedalWalker, this is a strong evidence that the diversity notation discussed in this paper truly leads to a better sample complexity even when Deep RL is employed.

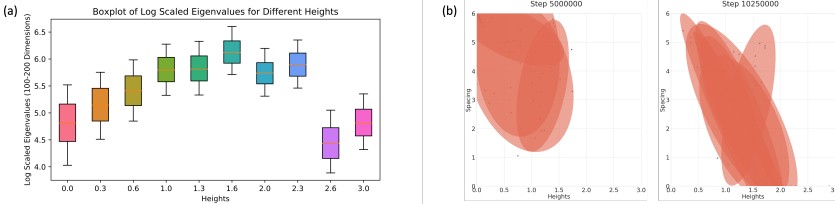

Figure 2: **(a)** Boxplot of the log-scaled eigenvalues of sample covariance matrices of the trained embeddings generated by the near optimal policies for different environments. **(b)** Task preference of automatically generated curriculum at 5M and 10M training steps respectively. The red regions are the regions where a task has a higher probability to be sampled.

## 7  DISCUSSIONS

In this paper, we propose a new perspective to understand the sample efficiency of myopic exploration design through diverse multitask learning. We show that by learning a diverse set of tasks, a multitask RL algorithm with myopic exploration design can be sample-efficient. This paper is a promising first step towards understanding the exploration benefits of MTRL.

**Towards diversity for general function classes.** Despite the rich discussion on explicit form of diversity set for tabular and linear MDP cases, how to achieve diversity for any general function class is an open problem. Recalling our proof for the Linear MDP case, a sufficient condition is to include a set of MDPs for each step $h$, such that the state distribution generated by their optimal policies satisfy the concentrability assumptions. In other words, any MDP with positive reward only at the step $h+1$ can be offline-learned through the dataset collected by these optimal policies. The diversity for general function classes poses the question on the number of tasks it takes to have sufficient coverage at the each step. We give a more detailed discussion of this topic in Appendix H.

**Improving sample-complexity bound.** Our sample complexity bound can be sub-optimal. For instance, Theorem 1 specialized to the tabular case has an upper sample complexity bound of $|\mathcal{M}|^2 S^3 H^5 A^2/\beta^2$, and the current optimal bound is $|\mathcal{M}|SA/\beta^2$ if tasks are learned independently. We conjecture that this gap may originate from two factors. First, the nature of the myopic exploration makes it less efficient because the exploration are conducted in a layered manner. Second, our algorithm collects trajectories for every MDP with the mixture of all the policy in each round, which may be improved if a curriculum is known beforehand.

## 8 ACKNOWLEDGEMENT

Prof. Tewari acknowledges the support of NSF via grant IIS-2007055. Prof. Stone runs the Learning Agents Research Group (LARG) at UT Austin. LARG research is supported in part by NSF (FAIN-2019844, NRT-2125858), ONR (N00014-18-2243), ARO (E2061621), Bosch, Lockheed Martin, and UT Austin's Good Systems grand challenge. Prof. Stone also serves as the Executive Director of Sony AI America and receives financial compensation for this work. The terms of this arrangement have been reviewed and approved by the University of Texas at Austin in accordance with its policy on objectivity in research.

Part of this work is completed while Dr. Ziping Xu is a postdoc at Harvard University. He is supported by Prof. Susan A. Murphy through NIH/NIDA P50DA054039, NIH/NIBIB, OD P41EB028242 and NIH/NIDCR UH3DE028723.

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
