## A    RELATED WORKS

**Multitask RL.**    Many recent theoretical works have contributed to understanding the benefits of MTRL (Agarwal et al., 2022; Brunskill & Li, 2013; Calandriello et al., 2014; Cheng et al., 2022; Lu et al., 2021; Uehara et al., 2021; Yang et al., 2022a; Zhang & Wang, 2021) by exploiting the shared structures across tasks. An earlier line of works (Brunskill & Li, 2013) assumes that tasks are clustered and the algorithm adaptively learns the identity of each task, which allows it to pool observations. For linear Markov Decision Process (MDP) settings (Jin et al., 2020b), Lu et al. (Lu et al., 2021) shows a bound on the sub-optimality of the learned policy by assuming a full-rank least-square value iteration weight matrix from source tasks. Agarwal et al. (Agarwal et al., 2022) makes a different assumption that the target transition probability is a linear combination of the source ones, and the feature extractor is shared by all the tasks. Our work differs from all these works as we focus on the reduced complexity of exploration design.

**Curriculum learning.**    Curriculum learning refers to adaptively selecting tasks in a specific order to improve the learning performance (Bengio et al., 2009) under a multitask learning setting. Numerous studies have demonstrated improved performance in different applications (Jiang et al., 2015; Pentina et al., 2015; Graves et al., 2017; Wang et al., 2021). However, theoretical understanding of curriculum learning remains limited. Xu & Tewari (2022) study the statistical benefits of curriculum learning under Supervised Learning setting. For RL, Li et al. (2022b) makes a first step towards the understanding of sample complexity gains of curriculum learning without an explicit exploration bonus, which is a similar statement as we make in this paper. However, their results are under strong assumptions, such as prior knowledge on the curriculum and a specific contextual RL setting with Lipschitz reward functions. This work can be seen as a more comprehensive framework of such benefits, where we discuss general MDPs with function approximation.

**Myopic exploration.**    Myopic exploration, characterized by its ease of implementation and effectiveness in many problems (Kalashnikov et al., 2018; Mnih et al., 2015), is the most commonly used exploration strategy. Many theory works (Dabney et al., 2020; Dann et al., 2022; Liu & Brunskill, 2018; Simchowitz & Foster, 2020) have discussed the conditions, under which myopic exploration is efficient. However, all these studies consider a single MDP and require strong conditions on the underlying environment. Our paper closely follows Dann et al. (2022) where they define Myopic Exploration Gap. An MDP with low Myopic Exploration Gap can be efficiently learned by exploration exploration.

## B    A FORMAL DISCUSSION ON CURRICULUM LEARNING

We formally discuss that how our theory could provide a potential explanation on the success of curriculum learning in RL (Narvekar et al., 2020). Although Algorithm 1 does not explicitly implement curriculum learning by ordering tasks, we argue that if any curriculum learn leads to polynomial sample complexity $\mathcal{C}$, then Algorithm 1 has $|\mathcal{M}|^2\mathcal{C}$ sample complexity. We denote a curricula by $((M_i, T_i))_{i=1}^T$ and an online algorithm that learns through the curricula interacts with $M_i$ for $T_i$ rounds by rolling out trajectories with the estimated optimal policy of $M_{i-1}$ with epsilon greedy. This curricula is implicitly included in Algorithm 1 with $|\mathcal{M}| \sum_i T_i$ rounds. To see this, let us say in phase $i$, the algorithm has mastered all tasks $M_1, \ldots, M_{i-1}$. Then by running Algorithm 1 $|\mathcal{M}|^2 T_i$ rounds, we will roll out $T_i$ trajectories on $M_i$ using the exploratory policy from $M_{i-1}$ on average, which reflects the schedules from curricula. This means that the sample-complexity of Algorithm 1 provides an upper bound on the sample complexity of underlying optimal curricula and in this way our theory provides some insights on the success of the curriculum learning.

## C    COMPARING SINGLE-TASK MEG AND MULTITASK MEG

**Proposition 1.**    *Let $\mathcal{M}$ be any set of MDPs and $\mathcal{F}$ be any function class. We have that $\alpha(f, \mathcal{M}, \mathcal{F}) \geq \alpha(f_M, \{M\}, \mathcal{F})/\sqrt{|\mathcal{M}|}$ for all $f \in (\mathcal{F})^{\otimes|\mathcal{M}|}$ and $M \in \mathcal{M}$.*

*Proof.* The proof is straightforward from the definition of multitask MEG. For any MDP $M$ and any $f \in \mathcal{F}^{\otimes |\mathcal{M}|}$, $\alpha(f_M, \{M\}, \mathcal{F})$ is the value to the following optimization problem

$$\sup_{\tilde{f} \in \mathcal{F}, c \geq 1} \frac{1}{\sqrt{c}} (V_{1,M}^{\tilde{f}} - V_{1,M}^{f_M}), \text{ s.t. for all } f' \in \mathcal{F} \text{ and } h \in [H],$$

$$\mathbb{E}_{\pi \tilde{f}}^{M} [(\mathcal{E}_h^M f')(s_h, a_h)]^2 \leq c \mathbb{E}_{\text{expl}(f)}^{M} [(\mathcal{E}_h^M f')(s_h, a_h)]^2$$

$$\mathbb{E}_{\pi f_M}^{M} [(\mathcal{E}_h^M f')(s_h, a_h)]^2 \leq c \mathbb{E}_{\text{expl}(f)}^{M} [(\mathcal{E}_h^M f')(s_h, a_h)]^2.$$

By choosing $c$ in Definition 3 by $c|\mathcal{M}|$, and $f'$ by the same $f'$ that attains the maximization in Single-task MEG, we have $\alpha(f, \mathcal{M}, \mathcal{F}) \geq \alpha(f_M, \{M\}, \mathcal{F})/\sqrt{|\mathcal{M}|}$ □

## D  EFFICIENT MYOPIC EXPLORATION FOR DETERMINISTIC MDP WITH KNOWN CURRICULUM

In light of the intrinsic connection between Algorithm 1 and curriculum learning. We present an interesting results for curriculum learning showing that any deterministic MDP can be efficiently learned through myopic exploration when a proper curriculum is given.

**Proposition 4.** *For any deterministic MDP $M$, with sparse reward, there exists a sequence of deterministic MDPs $M_1, M_2, \ldots, M_H$, such that the following learning process returns a optimal policy for $M$:*

1. *Initialize $\pi_0$ by a random policy.*

2. *For $t = 1, \ldots, n$, follow $\pi_{t-1}$ with an $\epsilon$-greedy exploration to collect $4At \log(H/\delta)$ trajectories denoted by $\mathcal{D}_t$. Compute the optimal policy $\pi_t$ from the model learned by $\mathcal{D}_t$.*

3. *Output $\pi_H$.*

*The above procedure will end in $O(AH^2 \log(H/\delta))$ episodes and with a probability at least $1 - \delta$, $\pi_H$ is the optimal policy for $M$.*

*Proof.* We construct the sequence in the following manner. Let the optimal policy for an MDP $M$ be $\pi_M^*$. Let the trajectory induced by $\pi_M^*$ be $\{s_0^*, a_0^*, \ldots, s_H^*, a_H^*\}$. The MDP $M$ receives a positive reward only when it reaches $s_H^*$. Without loss of generality, we assume that $M$ is initialized at a fixed state $s_0$. We choose $M_n$ such that

$$R_{M_i}(s, a) = \mathbb{1}(P_{M_n}(s, a) = s_i^*).$$

Furthermore, we set

$$P_{M_i}(s_i^* | s_i^*, a) = 1 \, \forall a \in \mathcal{A}$$

and

$$P_{M_i} = P_M$$

otherwise.

This ensures that any policy that reaches $s_i^*$ on the $i$'th step is an optimal policy.

We first provide an upper bound on the expected number of episodes for finding an optimal policy using the above algorithm for $M_i$.

Fix $2 \leq i \leq H$. Let $\epsilon = \frac{1}{i}$. Define $k = |A|$. Then

Then the probability for reaching optimal reward for $M_i$ is less than or equal to

$$(1 - \frac{1}{i})^{i-1}(\frac{1}{ki})$$

So the expected number of episodes to reach this optimal reward (and thus find an optimal policy) is

$$\frac{1}{(1 - \frac{1}{i})^{i-1}(\frac{1}{ki})} = (i-1)k(\frac{i}{i-1})^i \leq 4k(i-1)$$

since $i \geq 2$ and $(\frac{i}{i-1})^i$ is decreasing. By Chebyshev's inequality, a successful visit can be found in $4k(i-1)\log(H/\delta)$ with a probability at least $1 - \delta/H$.

The expected total number of episodes for the all the MDP's is therefore

$$\sum_{j=2}^{H} 4k(j-1)\log(H/\delta) \leq \frac{H}{2}(4kH)\log(H/\delta)$$

which is $O(kH^2 \log(H/\delta))$. $\qquad\square$

## E  GENERIC UPPER BOUND FOR SAMPLE COMPLEXITY

In this section, we prove the generic upper bound on sample complexity in Theorem 1. We first prove Lemma 2, which holds under the same condition of Theorem 1.

**Lemma 2.** *Consider a multitask RL problem with MDP set $\mathcal{M}$ and value function class $\mathcal{F}$ such that $\mathcal{M}$ is $(\tilde{\alpha}, \tilde{c})$-diverse. Then Algorithm 1 running $T$ rounds with exploration function* expl *satisfies that with a probability at least $1 - \delta$, the total number of rounds, where there exists an MDP $M$, such that $\pi^{\hat{f}_{t,M}}$ is $\beta$-suboptimal for $M$, can be upper bounded by*

$$\mathcal{O}\left(|\mathcal{M}|H^2 d_{BE}(\mathcal{F}, \Pi_{\mathcal{F}}, 1/\sqrt{T})\frac{\ln \tilde{c}(\beta)}{\tilde{\alpha}(\beta)^2}\ln\left(\frac{N'_{\mathcal{F}}\left(T^{-1}\right)\ln T}{\delta}\right)\right),$$

*where $\dim_{BE}(\mathcal{F}, \Pi_{\mathcal{F}}, 1/\sqrt{T})$ is the Bellman-Eluder dimension of class $\mathcal{F}$ and $N'_{\mathcal{F}}(\rho) = \sum_{h=1}^{H-1} N_{\mathcal{F}_h}(\rho)N_{\mathcal{F}_{h+1}}(\rho)$.*

*Proof.* Let us partition $\mathcal{F}_\beta$ into $\mathcal{F}_\beta = \{\mathcal{F}_{M,i}\}_{M \in \mathcal{M}, i \in [i_{\max}]}$ such that

$$\mathcal{F}_{M,i} \coloneqq \{f \in \mathcal{F}_\beta : c(f, \mathcal{M}, \mathcal{F}) \in [e^{i-1}, e^i] \text{ and } M(f, \mathcal{M}, \mathcal{F}) = M\}.$$

Furthermore, denote $(\hat{f}_{t,M})_{M \in \mathcal{M}}$ by $\hat{f}_t$. We define $\mathcal{K}_{M,i,t} = \{\tau \in [t], \hat{f}_\tau \in \mathcal{F}_{M,i}\}$. The proof in Dann et al. (2022) can be seen as bounding the sum of $\mathcal{K}_{M,i,t}$ for a specific $M$, while apply the same bound for each $M$, which leads to an extra $|\mathcal{M}|$ factor.

**Lemma 3.** *Under the same condition in Theorem 1 and the above definition, we have*

$$|\mathcal{K}_{M,i,T}| \leq \mathcal{O}\left(\frac{H^2 d_{BE}(\mathcal{F}, \Pi_{\mathcal{F}}, 1/\sqrt{T})}{\tilde{\alpha}(\beta)^2}\ln\frac{N'_{\mathcal{F}}(1/T)\ln(T)}{\delta}\right).$$

*Proof.* In the following proof, we fix an MDP $M$ and without further specification, the policies or rewards are with respect to the specific $M$. We study all the steps $t \in \mathcal{K}_{M,i,T}$.

For each $t \in \mathcal{K}_{M,i,T}$,

1. Recall that $\hat{\pi}_t$ is the mixture of exploration policy for all the MDPs: $\text{Mixture}(\{\text{expl}(\hat{\pi}_{t,M'})\}_{M' \in \mathcal{M}})$;

2. Define $\pi'_t$ as the improved policy that attains the maximum in the multitask myopic exploration gap for $\hat{f}_t$ in Definition 3.

Note that $\pi'_t$ is a policy for $M$ since $t \in \mathcal{K}_{M,i,t}$. A key step in our proof is to upper bound the difference between the value of the current policy and the value of $\pi'_t$. By Lemma 4, The total difference in return between the greedy policies and the improved policies can be bounded by

$$\sum_{t \in \mathcal{K}_{M,i,T}} (V_{1,M}^{\pi'_t}(s_1) - V_{1,M}^{\hat{\pi}_{t,M}}(s_1)) \leq \sum_{t \in \mathcal{K}_{M,i,T}}\sum_{h=1}^{H} \mathbb{E}_{\hat{\pi}_{t,M}}^{M}[(\mathcal{E}_h^M \hat{f}_{t,M})(s_h, a_h)] - \sum_{t \in \mathcal{K}_{M,i,T}}\sum_{h=1}^{H} \mathbb{E}_{\pi'_t}^{M}[(\mathcal{E}_h^M \hat{f}_{t,M})(s_h, a_h)],$$

$$\tag{3}$$

where the exportation is taken over the randomness of the trajectory sampled for MDP $M$.

Under the completeness assumption in Assumption 1, by Lemma 5 we show that with a probability $1 - \delta$ for all $(h, t) \in [H] \times [T]$,

$$\sum_{\tau=1}^{t-1} \mathbb{E}_{\hat{\pi}_\tau}^M \left[ (\mathcal{E}_h f_{t,M}) (s_h, a_h) \right]^2 \leq 3 \frac{t-1}{T} + 176 \ln \frac{6 N'_{\mathcal{F}}(1/T) \ln(2t)}{\delta}.$$

We consider only the event, where this condition holds. Since $c(\hat{f}_t, \mathcal{M}, \mathcal{F}) \leq e^i$ for all $t \in \mathcal{K}_{M,i,T}$, by Definition 3 we bound

$$\sum_{\tau \in \mathcal{K}_{M,i,t-1}} \mathbb{E}_{\pi'_\tau}^M \left[ (\mathcal{E}_h^M \hat{f}_{t,M}) (s_h, a_h) \right]^2$$

$$\leq \sum_{\tau \in [t-1]} \mathbb{E}_{\pi'_\tau}^M \left[ (\mathcal{E}_h^M \hat{f}_{t,M}) (s_h, a_h) \right]^2$$

$$\leq e^i \sum_{\tau \in [t-1]} \mathbb{E}_{\hat{\pi}_\tau}^M \left[ (\mathcal{E}_h^M \hat{f}_{t,M}) (s_h, a_h) \right]^2$$

$$\leq 179 e^i \ln \frac{6 N'_{\mathcal{F}}(1/T) \ln(2t)}{\delta}.$$

Combined with the distributional Eluder dimension machinery in Lemma 7, this implies that

$$\sum_{t \in \mathcal{K}_{M,i,T}} \left| \mathbb{E}_{\pi'_t}^M \left[ (\mathcal{E}_h^M \hat{f}_{t,M}) (s_h, a_h) \right] \right| \leq \mathcal{O} \left( \sqrt{e^i d_{BE}(\mathcal{F}, \Pi_{\mathcal{F}}, 1/\sqrt{T}) \ln \frac{N'_{\mathcal{F}}(1/T) \ln(T)}{\delta} |\mathcal{K}_{M,i,T}|} \right.$$

$$\left. + \min \left\{ |\mathcal{K}_{M,i,T}|, d_{BE}(\mathcal{F}, \Pi_{\mathcal{F}}, 1/\sqrt{T}) \right\} \right),$$

Note that we can derive the same upper-bound for $\sum_{t \in \mathcal{K}_{M,i,T}} \left| \mathbb{E}_{\pi_t}^M \left[ (\mathcal{E}_h^M \hat{f}_{t,M}) (s_h, a_h) \right] \right|$. Then plugging the above two bounds into Equation (3), we obtain

$$\sum_{t \in \mathcal{K}_{M,i,T}} (V_{1,M}^{\pi'_t}(s_1) - V_{1,M}^{\hat{\pi}_{t,M}}(s_1)) \leq \mathcal{O} \left( \sqrt{e^i H^2 d_{BE}(\mathcal{F}, \Pi_{\mathcal{F}}, 1/\sqrt{T}) \ln \frac{N'_{\mathcal{F}}(1/T) \ln(T)}{\delta} |\mathcal{K}_{M,i,T}|} + H d(\mathcal{F}'_i) \right).$$

By the definition of myopic exploration gap, we lower bound the LHS by

$$\sum_{t \in \mathcal{K}_{M,i,T}} (V_{1,M}^{\pi'_t}(s_1) - V_{1,M}^{\hat{\pi}_{t,M}}(s_1)) \geq |\mathcal{K}_{M,i,T}| \sqrt{e^{i-1}} \alpha_\beta.$$

Combining both bounds and rearranging yields

$$|\mathcal{K}_{M,i,T}| \leq \mathcal{O} \left( \frac{H^2 d_{BE}(\mathcal{F}, \Pi_{\mathcal{F}}, 1/\sqrt{T})}{\alpha_\beta^2} \ln \frac{N'_{\mathcal{F}}(1/T) \ln(T)}{\delta} \right).$$

$\square$

Summing over $M \in \mathcal{M}$ and $i \leq i_{max} < \ln \tilde{c}(\beta)$, we conclude Lemma 2. $\square$

To convert Lemma 2 into a sample complexity bound in Theorem 1, we show that for all $M$, $\hat{\pi}_M = \text{Mixture}(\{\pi^{\hat{f}_{t,M}}\})$ is $\beta$-optimal for $M$.

$$\max_\pi V_{1,M}^\pi - V_{1,M}^{\hat{\pi}_M}(s_1) = \mathcal{O} \left( \frac{\beta |\mathcal{M}| H^2 d_{BE}(\mathcal{F}, \Pi_{\mathcal{F}}, 1/\sqrt{T}) \frac{\ln \tilde{c}(\beta)}{\tilde{\alpha}(\beta)^2} \ln \left( \frac{N'_{\mathcal{F}}(T^{-1}) \ln T}{\delta} \right)}{T} \right).$$

To have the above suboptimality controlled at the level $\beta$, we will need

$$
\mathcal{O}\left(\frac{\beta|\mathcal{M}|H^2 d_{BE}(\mathcal{F},\Pi_{\mathcal{F}},1/\sqrt{T})\frac{\ln\tilde{c}(\beta)}{\tilde{\alpha}(\beta)^2}\ln\left(\frac{N'_{\mathcal{F}}\left(T^{-1}\right)\ln T}{\delta}\right)}{T}\right) = \beta.
$$

Assume that $d_{BE}(\mathcal{F},\Pi_{\mathcal{F}},\rho) = \mathcal{O}(d_{\mathcal{F}}^{\text{BE}}\log(1/\rho))$ and $\log(N'(\mathcal{F})(\rho)) = \mathcal{O}(d_{\mathcal{F}}^{\text{cover}}\log(1/\rho))$ by ignoring other factors, which holds for most regular function classes including tabular and linear classes (Russo & Van Roy, 2013), we have

$$
T = \mathcal{O}\left(|\mathcal{M}|H^2 d_{\mathcal{F}}^{BE} d_{\mathcal{F}}^{\text{cover}}\frac{\ln\tilde{c}(\beta)}{\tilde{\alpha}(\beta)^2}\ln\frac{1}{\delta}\ln(1/\rho)\right),
$$

where $\rho^{-1} = \mathcal{O}\left(|\mathcal{M}|H^2 d_{\mathcal{F}}^{\text{BE}} d_{\mathcal{F}}^{\text{cover}}\frac{\ln\tilde{c}(\beta)}{\tilde{\alpha}(\beta)^2}\ln(1/\delta)\right)$. The final bound has an extra $|\mathcal{M}|$ dependence because we execute a policy for each MDP in a round.

### E.1 Technical Lemmas

**Lemma 4 (Lemma 3 (Dann et al., 2022)).** *For any MDP $M$, let $f = \{f_h\}_{h\in[H]}$ with $f_h : \mathcal{S}\times\mathcal{A} \mapsto \mathbb{R}$ and $\pi^f$ is the greedy policy of $f$. Then for any policy $\pi'$,*

$$
V_1^{\pi'}(s_1) - V_1^{\pi^f}(s_1) \le \sum_{h=1}^{H}\mathbb{E}_{\pi^f}^M\left[(\mathcal{E}_h f)(s_h,a_h)\right] - \sum_{h=1}^{H}\mathbb{E}_{\pi'}^M\left[(\mathcal{E}_h f)(s_h,a_h)\right].
$$

**Lemma 5 (Modified from Lemma 4 (Dann et al., 2022)).** *Consider a sequence of policies $(\pi_t)_{t\in\mathbb{N}}$. At step $\tau$, we collect one episode using $\hat{\pi}_\tau$ and define $\hat{f}_\tau$ as the fitted Q-learning estimator up to step $t$ over the function class $\mathcal{F} = \{\mathcal{F}\}_{h\in[H]}$. Let $\rho \in \mathbb{R}^+$ and $\delta \in (0,1)$. If $\mathcal{F}$ satisfies Assumption 1, then with a probability at least $1 - \delta$, for all $h \in [H]$ and $t \in \mathbb{N}$,*

$$
\sum_{\tau=1}^{t-1}\mathbb{E}_{\hat{\pi}_\tau}^M[(\mathcal{E}_h\hat{f}_t)(s_h,a_h)]^2 \le 3\rho t + 176\ln\frac{6N'_{\mathcal{F}}(\rho)\ln(2t)}{\delta},
$$

*where $N'_{\mathcal{F}}(\rho) = \sum_{h=1}^{H} N_{\mathcal{F}_h}(\rho)N_{\mathcal{F}_{h+1}}(\rho)$ is the sum of $\ell_\infty$ covering number of $\mathcal{F}_h \times \mathcal{F}_{h+1}$ w.r.t. radius $\rho > 0$.*

*Proof.* The only difference between our statement and the statement in Dann et al. (2022) is that they consider $\hat{\pi}_\tau = \text{expl}(\hat{f}_\tau)$, while this statement holds for any data-collecting policy $\hat{\pi}_\tau$. To show this, we go through the complete proof here.

Consider a fixed $t \in \mathbb{N}$, $h \in [H]$ and $f = \{f_h, f_{h+1}\}$ with $f_h \in \mathcal{F}_h$, $f_{h+1} \in \mathcal{F}_{h+1}$. Let $(x_{t,h},a_{t,h},r_{t,h})_{t\in\mathbb{N},h\in[H]}$ be the collected trajectory in $[t]$. Then

$$
\begin{aligned}
Y_{t,h}(f) &= \left(f_h(x_{t,h},a_{t,h}) - r_{t,h} - \max_{a'}f_{h+1}(x_{t,h+1},a')\right)^2 - \left((\mathcal{T}_h f_{h+1})(x_{t,h},a_{t,h}) - r_{t,h} - \max_{a'}f_{h+1}(x_{t,h+1},a')\right)^2 \\
&= (f_h(x_{t,h},a_{t,h}) - (\mathcal{T}_h f_{h+1})(x_{t,h},a_{t,h})) \\
&\quad \times \left(f_h(x_{t,h},a_{t,h}) + (\mathcal{T}_h f_{h+1})(x_{t,h},a_{t,h}) - 2r_{t,h} - 2\max_{a'}f_{h+1}(x_{t,h+1},a')\right).
\end{aligned}
$$

Let $\mathfrak{F}_t$ be the $\sigma$-algebra under which all the random variables in the first $t-1$ episodes are measurable. Note that $|Y_{t,h}(f)| \le 4$ almost surely and the conditional expectation of $Y_{y,h}(f)$ can be written as

$$
\mathbb{E}\left[Y_{t,h}(f)\mid\mathfrak{F}_t\right] = \mathbb{E}\left[\mathbb{E}\left[Y_{t,h}(f)\mid\mathfrak{F}_t,x_{t,h},a_{t,h}\right]\mid\mathfrak{F}_t\right] = \mathbb{E}_{\pi_t}[(f_h - \mathcal{T}_h f_{h+1})(x_h,a_h)^2].
$$

The variance can be bounded by

$$
\text{Var}\left[Y_{t,h}(f)\mid\mathfrak{F}_t\right] \le \mathbb{E}\left[Y_{t,h}(f)^2\mid\mathfrak{F}_t\right] \le 16\mathbb{E}\left[(f_h - \mathcal{T}_h f_{h+1})(x_{t,h},a_{t,h})^2\mid\mathfrak{F}_t\right] = 16\mathbb{E}\left[Y_{t,h}(f)\mid\mathfrak{F}_t\right],
$$

where we used the fact that $|f_h(x_{t,h}, a_{t,h}) + (\mathcal{T}_h f_{h+1})(x_{t,h}, a_{t,h}) - 2r_{t,h} - 2\max_{a'} f_{h+1}(x_{h+1}, a')| \leq 4$ almost surely. Applying Lemma 6 to the random variable $Y_{t,h}(f)$, we have that with probability at least $1 - \delta$, for all $t \in \mathbb{N}$,

$$\sum_{i=1}^{t} \mathbb{E}\left[Y_{i,h}(f) \mid \mathfrak{F}_i\right] \leq 2A_t \sqrt{\sum_{i=1}^{t} \mathrm{Var}\left[Y_{i,h}(f) \mid \mathfrak{F}_i\right] + 12A_t^2} + \sum_{i=1}^{t} Y_{i,h}(f)$$

$$\leq 8A_t \sqrt{\sum_{i=1}^{t} \mathbb{E}\left[Y_{i,h}(f) \mid \mathfrak{F}_i\right] + 12A_t^2} + \sum_{i=1}^{t} Y_{i,h}(f),$$

where $A_t = \sqrt{2\ln\ln(2t) + \ln(6/\delta)}$. Using AM-GM inequality and rearranging terms in the above we have

$$\sum_{i=1}^{t} \mathbb{E}\left[Y_{i,h}(f) \mid \mathfrak{F}_i\right] \leq 2\sum_{i=1}^{t} Y_{i,h}(f) + 88A_t^2 \leq 2\sum_{i=1}^{t} Y_{i,h}(f) + 176\ln\frac{6\ln(2t)}{\delta}.$$

Let $\mathcal{Z}_{\rho,h}$ be a $\rho$-cover of $\mathcal{F}_h \times \mathcal{F}_{h+1}$. Now taking a union bound over all $\phi_h \in \mathcal{Z}_{\rho,h}$ and $h \in [H]$, we obtain that with probability at least $1 - \delta$ for all $\phi_h$ and $h \in [H]$

$$\sum_{i=1}^{t} \mathbb{E}\left[Y_{i,h}\left(\phi_h\right) \mid \mathfrak{F}_i\right] \leq 2\sum_{i=1}^{t} Y_{i,h}\left(\phi_h\right) + 176\ln\frac{6N_{\mathcal{F}}'(\rho)\ln(2t)}{\delta}.$$

This implies that with probability at least $1 - \delta$ for all $f = \{f_h, f_{h+1}\} \in \mathcal{F}_h \times \mathcal{F}_{h+1}$ and $h \in [H]$,

$$\sum_{i=1}^{t} \mathbb{E}\left[Y_{i,h}(f) \mid \mathfrak{F}_i\right] \leq 2\sum_{i=1}^{t} Y_{i,h}(f) + 3\rho(t-1) + 176\ln\frac{6N_{\mathcal{F}}'(\rho)\ln(2t)}{\delta}.$$

Let $\hat{f}_{t,h}$ be the $h$-th component of the function $\hat{f}_t$. The above inequality holds in particular for $f = \{\hat{f}_{t,h}, \hat{f}_{t,h+1}\}$ for all $t \in \mathbb{N}$. Finally, we have

$$\sum_{i=1}^{t-1} Y_{i,h}\left(\hat{f}_t\right) = \sum_{i=1}^{t-1}\left(\hat{f}_{t,h}\left(s_{i,h}, a_{i,h}\right) - r_{i,h} - \max_{a'}\hat{f}_{t,h+1}\left(s_{i,h+1}, a'\right)\right)^2$$

$$- \sum_{i=1}^{t-1}\left(\left(\mathcal{T}_h\hat{f}_{t,h+1}\right)\left(s_{i,h}, a_{i,h}\right) - r_{i,h} - \max_{a'}\hat{f}_{t,h+1}\left(s_{i,h+1}, a'\right)\right)^2$$

$$= \inf_{f' \in \mathcal{F}_h}\sum_{i=1}^{t-1}\left(f'\left(s_{i,h}, a_{i,h}\right) - r_{i,h} - \max_{a'}\hat{f}_{t,h+1}\left(s_{i,h+1}, a'\right)\right)^2$$

$$- \sum_{i=1}^{t-1}\left(\left(\mathcal{T}_h\hat{f}_{t,h+1}\right)\left(s_{i,h}, a_{i,h}\right) - r_{i,h} - \max_{a'}\hat{f}_{t,h+1}\left(s_{i,h+1}, a'\right)\right)^2$$

$$\leq 0,$$

where the last inequality follows from the completeness in Assumption 1. $\qquad\square$

**Lemma 6** (Time-Uniform Freedman Inequality). *Suppose $\{X_t\}_{t=1}^{\infty}$ is a martingale difference sequence with $|X_t| \leq b$. Let*

$$\mathrm{Var}_\ell\left(X_\ell\right) = \mathrm{Var}\left(X_\ell \mid X_1, \cdots, X_{\ell-1}\right).$$

*Let $V_t = \sum_{\ell=1}^{t} \mathrm{Var}_\ell(X_\ell)$ be the sum of conditional variances of $X_t$. Then we have that for any $\delta' \in (0, 1)$ and $t \in \mathbb{N}$*

$$\mathbb{P}\left(\sum_{\ell=1}^{t} X_\ell > 2\sqrt{V_t}A_t + 3bA_t^2\right) \leq \delta'$$

*where $A_t = \sqrt{2\ln\ln(2(\max(V_t/b^2, 1))) + \ln(6/\delta')}$.*

**Lemma 7** (Lemma 41 (Jin et al., 2021a)). *Given a function class $\Phi$ defined on $\mathcal{X}$ with $|\phi(x)| \leq C$ for all $(\phi, x) \in \Phi \times \mathcal{X}$ and a family of probability measures $\Pi$ over $\mathcal{X}$. Suppose sequences $\{\phi_i\}_{i \in [K]} \subset \Phi$ and $\{\mu_i\}_{i \in [K]} \subset \Pi$ satisfy for all $k \in [K]$ that $\sum_{i=1}^{k-1} (\mathbb{E}_{\mu_i}[\phi_k])^2 \leq \beta$. Then for all $k \in [K]$ and $w > 0$,*

$$\sum_{t=1}^{k} |\mathbb{E}_{\mu_t}[\phi_t]| \leq O\left(\sqrt{\dim_{DE}(\Phi, \Pi, \omega)\beta k} + \min\{k, \dim_{DE}(\Phi, \Pi, \omega)\}C + k\omega\right).$$

**Proof of Theorem 1.** Denote $B = \Theta\left(|\mathcal{M}|^2 H^2 d_{\text{BE}} \frac{\ln \tilde{c}(\beta)}{\tilde{\alpha}(\beta)^2 \beta} \ln\left(\frac{\bar{N}_{\mathcal{F}}\left(T^{-1}\right)\ln T}{\delta}\right)\right)$. The following Corollary transform Lemma 2 to Theorem 1, whose proof directly follows by taking $T = B/\beta$. Since at most $B$ rounds are suboptimal according to Lemma 2, the mixing of all $T$ policies are $\beta$-optimal. This leads to a sample complexity

$$\mathcal{C}(\tilde{\alpha}, \tilde{c}) = \Theta\left(|\mathcal{M}|^2 H^2 d_{\text{BE}} \frac{\ln \tilde{c}(\beta)}{\tilde{\alpha}(\beta)^2 \beta} \ln\left(\frac{\bar{N}_{\mathcal{F}}\left(T^{-1}\right)\ln T}{\delta}\right)\right)$$

# F    OMITTED PROOFS FOR CASE STUDIES

## F.1    LINEAR MDP CASE

Note that in this section, we use $\mathbb{E}_\pi$ for the expectation over transition w.r.t a policy $\pi$.

**Lemma 8.** *Let $\mathcal{F}$ be the function class in Proposition 3. For any policy $\pi$ such that $\lambda_{min}(\Phi_h^\pi) \geq \underline{\lambda}$, then for any policy $\pi'$ and $f' \in \mathcal{F}$, $\mathbb{E}_{\pi'}\left[\left(\mathcal{E}_h^2 f'\right)(s_h, a_h)\right] \leq \mathbb{E}_\pi\left[\left(\mathcal{E}_h^2 f'\right)(s_h, a_h)\right]/\underline{\lambda}$.*

*Proof.* Recall that $\Phi_h^\pi \coloneqq \mathbb{E}_\pi \phi_h(s_h, a_h)\phi_h(s_h, a_h)^\top$.

We derive the Bellman error term using the fact that $f'$ is a linear function and the transitions admit the linear function as well. For any policy $\pi$, we have

$$\mathbb{E}_\pi[(\mathcal{E}_h^2 f')(s_h, a_h)]$$

$$= \mathbb{E}_\pi\left[\left(f_h'(s_h, a_h) - \phi_h(s_h, a_h)^\top \theta_h - \max_{a'} \mathbb{E}_{s_{h+1}}[f_{h+1}'(s_{h+1}, a') \mid s_h, a_h]\right)^2\right]$$

$$= \mathbb{E}_\pi\left[\left(\phi_h(s_h, a_h)^\top w_h - \phi_h(s_h, a_h)^\top \theta_h - \max_{a'} \mathbb{E}_{s_{h+1}}[\phi_{h+1}(s_{h+1}, a')^\top w_{h+1} \mid s_h, a_h]\right)^2\right]$$

$$= \mathbb{E}_\pi\left[\left(\phi_h(s_h, a_h)^\top w_h - \phi_h(s_h, a_h)^\top \theta_h - \phi_h(s_h, a_h)^\top \int_{s'} \phi_{h+1}(s', \pi_{h+1}^{f'}(s'))^\top w_{h+1}\mu_h(s')ds'\right)^2\right]$$

$$= \mathbb{E}_\pi\left[\left(\phi_h(s_h, a_h)^\top (w_h - \theta_h - w_{h+1}')\right)^2\right]$$

$$= (w_h - \theta_h - w_{h+1}')^\top \mathbb{E}_\pi\left[\phi_h(s_h, a_h)\phi_h(s_h, a_h)^\top\right](w_h - \theta_h - w_{h+1}')$$

where $w_{h+1}' = \int_{s'} \phi_{h+1}(s', \pi_{h+1}^{f'}(s'))^\top w_{h+1}\mu_h(s')ds'$. Since by the assumption in Definition 6 that $\|\phi_h(s, a)\| \leq 1$ for any $s, a$, we have $\Phi_h^{\pi'} \prec I$. The result follow by the condition that $\lambda_{\min}(\Phi_h^\pi) \geq \underline{\lambda}$. $\square$

**Lemma 1.** *Fix a step $h$. Let $\{M_{i,h}\}_{i \in [d]}$ be the $d$ MDPs such that $\theta_{h, M_{i,h}} = e_i$ as in Definition 6. Let $\{\pi_i\}_{i=1}^d$ be $d$ policies such that $\pi_i$ is a $\beta$-optimal policy for $M_{i,h}$ with $\beta < b_1/2$. Let $\tilde{\pi} = \text{Mixture}(\{\text{expl}(\pi_i)\}_{i=1}^d)$. Then for any $\nu \in \mathbb{S}^{d-1}$, we have $\lambda_{\min}(\Phi_{h+1}^{\tilde{\pi}}) \geq \epsilon_h \prod_{h'=1}^{h-1}(1 - \epsilon_{h'})b_1^2/(2dA)$.*

*Proof.* Let $\pi$ be any stationary policy and recall that $\Pi$ is the set of all the stationary policies. We denote $A_h^\pi(s') \sim \pi_h(s')$ by the random variable for the action sampled at the step $h$ using policy $\pi$ given the state is $s'$. Let $\phi_h^\pi \coloneqq \mathbb{E}_\pi \phi_h(s_h, a_h)$.

We further define
$$a_{h+1}^{\nu}(s) := \arg\max_{a \in \mathcal{A}}[\nu^{\top}\phi_{h+1}(s,a)\phi_{h+1}(s,a)^{\top}\nu].$$

Lower bound the following quadratic term for any unit vector $\nu \in \mathbb{R}^d$,

$$\max_{\pi \in \Pi}\nu^{\top}\Phi_{h+1}^{\pi}\nu$$
$$= \max_{\pi \in \Pi}\mathbb{E}_{\pi}\left[\int_{s'}\nu^{\top}\phi_{h+1}(s',A_{h+1}^{\pi}(s'))\phi_{h+1}(s',A_{h+1}^{\pi}(s'))^{\top}\nu\mu_h(s')^{\top}\phi_h(s_h,a_h)ds'\right]$$
$$= \max_{\pi}\mathbb{E}_{\pi}[\phi_h(s_h,a_h)^{\top}]\left(\int_{s'}\nu^{\top}\phi_{h+1}(s',a_{h+1}^{\nu}(s'))\phi_{h+1}(s',a_{h+1}^{\nu}(s'))^{\top}\nu\mu_h(s')ds'\right)$$
$$= \max_{\pi \in \Pi}(\phi_h^{\pi})^{\top}w_{h+1}^{\nu}.$$

where we let $w_{h+1}^{\nu} := \int_{s'}\nu^{\top}\phi_{h+1}(s',a_{h+1}^{\nu}(s'))\phi_{h+1}(s',a_{h+1}^{\nu}(s'))^{\top}\nu\mu_h(s')^{\top}ds'$.

By Assumption 2, we have $\max_{\pi \in \Pi}\mathbb{E}_{\pi}[\phi_h(s_h,a_h)^{\top}]w_{h+1}^{\nu} \geq b_1^2$.

For the mixture policy $\tilde{\pi}$ defined in our lemma,

$$\nu^{\top}\Phi_{h+1}^{\tilde{\pi}}\nu = \frac{1}{d}\sum_{i=1}^{d}\mathbb{E}_{\mathrm{expl}(\pi_i)}[\nu^{\top}\phi_{h+1}(s_{h+1},a_{h+1})\phi_{h+1}(s_{h+1},a_{h+1})^{\top}\nu]$$
$$\geq \frac{\epsilon_h\prod_{h'=1}^{h-1}(1-\epsilon_{h'})}{Ad}\sum_{i=1}^{d}(\phi_h^{\pi_i})^{\top}w_{h+1}^{\nu}. \tag{4}$$

Since $\pi_i$ is a $b_1/2$-optimal policy for MDP $M_{i,h}$ and again by Assumption 2, we have

$$\theta_{h,M_{i,h}}^{\top}\phi_h^{\pi_i} \geq \frac{1}{2}\max_{\pi \in \Pi}\theta_{h,M_{i,h}}^{\top}\phi_h^{\pi}. \tag{5}$$

For any vector $\nu \in \mathbb{R}^d$, let $[\nu]_i$ be the $i$-th dimension of the vector. Note that $\theta_{h,M_{i,h}} = e_i$, (5) indicates $[\phi_h^{\pi_i}]_i \geq \frac{1}{2}\max_{\pi}[\phi_h^{\pi}]_i$.

Combining the inequality (5) with (4), we have

$$\nu^{\top}\Phi_{h+1}^{\tilde{\pi}}\nu = \frac{\epsilon_h\prod_{h'=1}^{h-1}(1-\epsilon_{h'})}{dA}\sum_{i=1}^{d}\sum_{j=1}^{d}[\phi_h^{\pi_i}]_j[w_{h+1}^{\nu}]_j$$
$$\geq \frac{\epsilon_h\prod_{h'=1}^{h-1}(1-\epsilon_{h'})}{dA}\sum_{i=1}^{d}[\phi_h^{\pi_i}]_i[w_{h+1}^{\nu}]_i$$
$$\geq \frac{\epsilon_h\prod_{h'=1}^{h-1}(1-\epsilon_{h'})}{dA}\sum_{i=1}^{d}\max_{\pi}[\phi_h^{\pi}]_i[w_{h+1}^{\nu}]_i$$
$$\geq \frac{\epsilon_h\prod_{h'=1}^{h-1}(1-\epsilon_{h'})}{2dA}\max_{\pi}(\phi_h^{\pi})^{\top}w_{h+1}^{\nu}$$
$$\geq \frac{\epsilon_h\prod_{h'=1}^{h-1}(1-\epsilon_{h'})b_1^2}{2dA}$$

$\square$

## F.2 PROOF OF THEOREM 2

**Theorem 2.** *Consider $\mathcal{M}$ defined in Definition 7. With Assumption 2 holding and $\beta \leq b_1/2$, for any $f \in \mathcal{F}_{\beta}$, we have lower bound $\alpha(f,\mathcal{F},\mathcal{M}) \geq \sqrt{e\beta^2 b_1^2/(2A|\mathcal{M}|H)}$ by setting $\epsilon_h = 1/h$.*

*Proof.* Let $h'$ be the smallest $h$, such that there exists $M_{i,h}$, $\pi^{f_{M_{i,h}}}$ is $\beta$-suboptimal. Let $(i',h')$ be the index of the MDP that has the suboptimal policy. We show that $M_{i',h'}$ has lower bounded myopic exploration gap.

By definition, $f$ is $\beta$-optimal for any MDP $M_{i,h'-1}$. By Lemma 1, letting $\tilde{\pi} = \text{expl}(f, \epsilon_{h'})$, we have

$$\nu^\top \Phi_{h'+1}^{\tilde{\pi}} \nu \geq \frac{\epsilon_{h'} \prod_{h''=1}^{h'-1}(1-\epsilon_{h''})b_1^2}{2A|\mathcal{M}|}.$$

By Lemma 8, we have that the optimal value function $f^*$ for MDP $M_{i',h'}$ satisfies that for any $f'$

$$\mathbb{E}_{\pi^{f^*}}^M \left[ \left( \mathcal{E}_h^2 f' \right)(s_h, a_h) \right] \leq \frac{2A|\mathcal{M}|}{\epsilon_{h'} \prod_{h''=1}^{h'-1}(1-\epsilon_{h''})b_1^2} \mathbb{E}_\pi^M \left[ \left( \mathcal{E}_h^2 f' \right)(s_h, a_h) \right].$$

Thus, by Definition 3, the myopic exploration gap for $f$ is lower bounded by

$$\beta \frac{1}{\sqrt{c}} = \beta \sqrt{\frac{\epsilon_{h'} \prod_{h''=1}^{h'-1}(1-\epsilon_{h''})b_1^2}{2A|\mathcal{M}|}} \geq \sqrt{\frac{\beta^2 b_1^2}{2A|\mathcal{M}|eH}},$$

if we choose $\epsilon_h = 1/(h+1)$. $\qquad\square$

### F.3 LINEAR QUADRATIC REGULATOR

To demonstrate the generalizability of the proposed framework, we introduce another interesting setting called Linear Quadratic Regulator (LQR). LQR takes continuous state space $\mathbb{R}^{d_s}$ and action space $\mathbb{R}^{d_a}$. In the LQR system, the state $s_h \in \mathbb{R}^{d_s}$ evolves according to the following transition: $s_{h+1} = A_h s_h + B_h a_h$, where $A_h \in \mathbb{R}^{d_s \times d_s}$, $B_h \in \mathbb{R}^{d_s \times d_a}$ are unknown system matrices that are shared by all the MDPs. We denote $s_h = (s_h, a_h)$ as the state-action vector. The reward function for an MDP $M$ takes a known quadratic form $r_{h,M}(s,a) = s^\top R_{h,M}^s s + a^\top R_{h,M}^a a$, where $R_{h,M}^s \in \mathbb{R}^{d_s \times d_s}$ and $R_{h,M}^a \in \mathbb{R}^{d_a \times d_a}$ [4].

Note that LQR is more commonly studied for the infinite-horizon setting, where stabilizing the system is a primary concern of the problem. We consider the finite-horizon setting, which alleviates the difficulties on stabilization so that we can focus our discussion on exploration. Finite-horizon LQR also allows us to remain consistent notations with the rest of the paper. A related work (Simchowitz & Foster, 2020) states that naive exploration is optimal for online LQR with a condition that the system injects a random noise onto the state observation with a full rank covariance matrix $\Sigma \succ 0$. Though this is a common assumption in LQR literature, one may notice that the analog of this assumption in the tabular MDP is that any state and action pair has a non-zero probability of visiting any other state, which makes naive exploration sample-efficient trivially. In this section, we consider a deterministic system, where naive exploration does not perform well in general.

**Properties of LQR.** It can be shown that the optimal actions are linear transformations of the current state (Farjadnasab & Babazadeh, 2022; Li et al., 2022a).

The optimal linear response is characterized by the discrete-time Riccati equation given by

$$P_{h,M} = A_h^\top (P_{h+1,M} - P_{h+1,M}\bar{R}_{h+1,M}^{-1} B_h^\top P_{h+1,M})A_h + R_{h,M}^s,$$

where $\bar{R}_{h+1,M} = R_h^a + B_h^\top P_{h+1,M} A_h$ and $P_{H+1} = \mathbf{0}$. Assume that the solution for the above equation is $\{P_{h,M}^*\}_{h \in [H+1]}$, then the optimal control actions takes the form

$$a_h = F_{h,M}^* s_h, \text{ where } F_{h,M}^* = -(R_{h,M}^s + B_h^\top P_{h,M}^* B_h)^{-1} B^\top P_{h,M}^* A_h.$$

and optimal value function takes the quadratic form: $V_{h,M}^*(s) = s^\top P_{h,M}^* s$ and

$$Q_{h,M}^*(x) = x^\top \begin{bmatrix} R_{h,M}^s + A_h^\top P_{h+1,M}^* A_h & A_h^\top P_{h+1,M}^* B_h \\ B_h^\top P_{h+1,M}^* A_h & R_{h,M}^a + B_h^\top P_{h+1,M}^* B_h \end{bmatrix} x.$$

This observation allows us to consider the following function approximation

$$\mathcal{F} = (\mathcal{F}_h)_{h \in [H+1]}, \text{ where each } \mathcal{F}_h = \{x \mapsto x^\top G_h x : G_h \in \mathbb{R}^{(d_s+d_a) \times (d_s+d_a)}\}.$$

The quadratic function class satisfies Bellman realizability and completeness assumptions.

---

[4] Note that LQR system often consider a cost function and the goal of the agent is to minimize the cumulative cost with $R_{h,M}^s$ being semi-positive definite. We formulation this as a reward maximization problem for consistency. Thus, we consider $R_{h,M}^s \prec \mathbf{0}$

**Definition 8** (Diverse LQR Task Set). *Inspired by the task construction in linear MDP case, we construct the diverse LQR set by $\mathcal{M} = \{M_{i,h}\}_{i\in[d_s],h\in[H]}$ such that these MDPs all share the same transition matrices $A_h$ and $B_h$ and each $M_{i,h}$ has $R^s_{h',M_{i,h}} = \mathbb{1}[h' = h]e_i e_i^\top$ and $R^a_{h',M_{i,h}} = -I$.*

**Assumption 3** (Regularity parameters). *Given the task set in Definition 8, we define some constants that appears on our bound. Let $\pi^*_{i,h}$ be the optimal policy for $M_{i,h}$. Let*

$$b_4 = \max_{i,h} \mathbb{E}_{\pi^*_{i,h}} \max_{h'} \|s_{h'}\|_2, \text{ and } b_5 = \max_{i,h} \mathbb{E}_{\pi^*_{i,h}} \max_{h'} \|a_{h'}\|_2.$$

These regularity assumption is reasonable because the optimal actions are linear transformations of states and we consider a finite-horizon MDP, with $F^*_h$ having upper bounded eigenvalues.

Similarly to the linear MDP case, we assume that the system satisfies some visibility assumption.

**Assumption 4** (Coverage Assumption). *For any $\nu \in \mathbb{R}^{d_s-1}$, there exists a policy $\pi$ with $\|a_h\|_2 \leq 1$ such that*

$$\max_\pi \mathbb{E}_\pi[s_h^\top \nu] \geq b_3, \text{ for } b_3 > 1.$$

**Theorem 3.** *Given Assumption 3, 4 and the diverse LQR task set in Definition 8, we have that for any $f \in \mathcal{F}_\beta$ with $\beta \leq (b_3^2 - 1)b_5^2/2$,*

$$\alpha(f, \mathcal{F}, \mathcal{M}) = \Omega\left(\frac{\max\{b_4^2, b_5^2\}b_4^2}{d_s H \max\{(b_3^2 - 1)b_5^2, d_s\sigma^2\}(b_3^2 - 1)b_5^2}\right).$$

### F.4    PROOF OF THEOREM 3

**Lemma 9.** *Assume that we have a set of policies $\{\pi_i\}_{i\in[d]}$ such that the $i$-th policy is a $(b_3^2 - 1)b_5^2/2$-optimal policy for LQR with $R^s_{h,i} = e_i e_i^\top$ and $R^a_{h,i} = -I$. Let $\tilde{\pi} = \text{Mixture}(\text{expl}\{\pi_i\})$. Then we have*

$$\lambda_{min}(\mathbb{E}_{\tilde{\pi}} s_{h+1} s_{h+1}^\top) \geq \frac{d_s \max\{\underline{\lambda}, d\sigma^2\}}{2\max\{b_4^2, b_5^2\}\prod_{h'=1}^{h-1}(1 - \epsilon_{h'})\epsilon_h}\underline{\lambda},$$

*with $\underline{\lambda} = (b_3^2 - 1)b_5^2$.*

*Proof.* We directly analyze the state covariance matrix at the step $h + 1$. Let $\eta_h \sim \mathcal{N}(0, \sigma^2)$

$$\mathbb{E}_{\tilde{\pi}} s_{h+1} s_{h+1}^\top = \mathbb{E}_{\tilde{\pi}}(A_h s_h + B_h a_h)(A_h s_h + B_h a_h)^\top$$

$$\succeq \frac{\prod_{h'=1}^{h-1}(1 - \epsilon_{h'})\epsilon_h}{d_s} \sum_{i=1}^{d_s} \left(\mathbb{E}_{\pi_i}(A_h s_h + B_h \eta_h)(A_h s_h + B_h \eta_h)^\top\right)$$

$$= \frac{\prod_{h'=1}^{h-1}(1 - \epsilon_{h'})\epsilon_h}{d_s} \sum_{i=1}^{d_s} \left(A_h \mathbb{E}_{\pi_i} s_h s_h^\top A_h^\top + B_h \mathbb{E}\eta_h \eta_h^\top B_h^\top\right) \tag{6}$$

To proceed, we show that $\sum_{i=1}^{d_s} \mathbb{E}_{\pi_i} s_h s_h^\top \succeq \underline{\lambda}I$.

From Assumption 4, we have $\mathbb{E}_{\pi^*_i}[s_h^\top e_i e_i^\top s_h - a_h a_h^\top] \succeq b_3^2 b_5^2 - b_5^2$, and by the fact that $\pi_i$ is a $(b_3^2 - 1)b_5^2/2$-optimal policy, we have

$$\mathbb{E}_{\pi^*_i}[s_h^\top e_i e_i^\top s_h - a_h a_h^\top] \succeq (b_3^2 b_5^2 - b_5^2)/2.$$

Since $\mathbb{E}_{\pi_i} a_h a_h^\top \succeq 0$, we have $\mathbb{E}_{\pi_i}[s_h^\top e_i e_i^\top s_h] \succeq (b_3^2 - 1)b_5^2/2$. Therefore, $\sum_{i=1}^{d_s} \mathbb{E}_{\pi_i} s_h s_h^\top \succeq \underline{\lambda}I$ with $\underline{\lambda} = (b_3^2 - 1)b_5^2/2$.

Combined with (6), we have

$$\mathbb{E}_{\tilde{\pi}} s_{h+1} s_{h+1}^\top \succeq \frac{\prod_{h'=1}^{h-1}(1 - \epsilon_{h'})\epsilon_h}{d_s} \left(\underline{\lambda} A_h A_h^\top + d_s\sigma^2 B_h B_h^\top\right).$$

Apply Assumption 4 again, for each $\nu_i = e_i, i = 1, \ldots, d_s$, there exists some policy $\pi'_i$ with $\|a_h\|_2 \leq b_5$, such that $\nu_i^\top \mathbb{E}_{\pi'_i} s_{h+1} s_{h+1}^\top \nu_i \geq b_3^2 b_5^2 - b_5^2$. Therefore, we have that $\sum_{i=1}^{d_s} \mathbb{E}_{\pi'_i} s_{h+1} s_{h+1}^\top \succeq (b_3^2 - 1)b_5^2 I$

The proof is completed by

$$\sum_{i=1}^{d_s} \mathbb{E}_{\pi_i'} s_{h+1} s_{h+1}^\top \preceq 2 \sum_{i=1}^{d_s} \left( A_h \mathbb{E}_{\pi_i'} s_h s_h^\top A_h^\top + B_h \mathbb{E}_{\pi_i'} a_h a_h^\top B_h^\top \right)$$

$$\preceq 2 \sum_{i=1}^{d_s} \left( b_4^2 A_h A_h^\top + b_5^2 B_h \mathbb{E}_{\pi_i'} B_h^\top \right)$$

$$\preceq \frac{2 \max\{b_4^2, b_5^2\}}{\max\{\underline{\lambda}, d\sigma^2\}} \frac{\prod_{h'=1}^{h-1}(1 - \epsilon_{h'})\epsilon_h}{d_s} \mathbb{E}_{\tilde{\pi}} s_{h+1} s_{h+1}^\top.$$

To complete the proof of Theorem 3, we combine Lemma 10 and Lemma 9.

$\square$

## F.5 SUPPORTING LEMMAS

Lemma 10 shows that having a full rank covariance matrix for the state $s_h$ is a sufficient condition for bounded occupancy measure.

**Lemma 10.** *Let $\mathcal{F}$ be the function class described above. For any policy $\pi$ and $h$ such that*

$$\lambda_{min}(\mathbb{E}_\pi[s_h s_h^\top]) \geq \underline{\lambda},$$

*we have for any $\pi'$ such that $\max_h \|s_h\|_2 \leq b_4$, and for any $f' \in \mathcal{F}$,*

$$\mathbb{E}_{\pi'}^M \left[ \left( \mathcal{E}_h^2 f' \right)(s_h, a_h) \right] \leq \frac{b_4^2}{\underline{\lambda}^2} \mathbb{E}_\pi^M \left[ \left( \mathcal{E}_h^2 f' \right)(s_h, a_h) \right].$$

*Proof.* Lemma 11 shows that the Bellman error also takes a quadratic form of $s_h$.

**Lemma 11.** *For any $f \in \mathcal{F}$, there exists some matrix $\tilde{G}_h$ such that $(\mathcal{E}_h f)(x) = x^\top \tilde{G}_h x$.*

To complete the proof of Lemma 10, let $w_h = s_h \otimes s_h$ be the Kronecker product between $s_h$ and itself. By Lemma 11, we can write $(\mathcal{E}_h f)(s_h) = \text{Vec}(\tilde{G}_h)^\top w_h$. Again, this is an analog of the linear form we had for thee linear MDP case. Thus, we can write $(\mathcal{E}_h^2 f)(s_h) = \text{Vec}(\tilde{G}_h)^\top w_h w_h^\top \text{Vec}(\tilde{G}_h)$.

By Lemma 12 and the fact that $\mathbb{E}_\pi(w_h w_h^\top) = \mathbb{E}_\pi(s_h s_h^\top) \otimes \mathbb{E}_\pi(s_h s_h^\top)$, we have $\lambda_{\min}(\mathbb{E}_\pi w_h w_h^\top) \geq \underline{\lambda}^2$.

For any other policy $\pi'$, and using the fact that $\|w_h\| \leq b_4^2$, we have

$$\mathbb{E}_{\pi'}(\mathcal{E}_h^2 f)(s_h) = \mathbb{E}_{\pi'}[\text{Vec}(\tilde{G}_h)^\top w_h w_h^\top \text{Vec}(\tilde{G}_h)] \leq \frac{b_4^2}{\underline{\lambda}^2} \mathbb{E}_\pi[\text{Vec}(\tilde{G}_h)^\top w_h w_h^\top \text{Vec}(\tilde{G}_h)] \leq \frac{b_4^2}{\underline{\lambda}^2} \mathbb{E}_\pi(\mathcal{E}_h^2 f)(s_h).$$

$\square$

**Lemma 11.** *For any $f \in \mathcal{F}$, there exists some matrix $\tilde{G}_h$ such that $(\mathcal{E}_h f)(x) = x^\top \tilde{G}_h x$.*

*Proof.* The Bellman error of the LQR can be written as

$$(\mathcal{E}_h f)(x) = \left( x^\top G_h x - s^\top R_h^s s - a^\top R_h^a a - \max_{a' \in \mathbb{R}^{d_a}} [(A_h s + B_h a)^\top, a'^\top] G_{h+1} \begin{bmatrix} A_h s + B_h a \\ a' \end{bmatrix} \right)$$

Note that the optimal $a'$ can be written as some linear transformation of $x$. Thus we can write

$$\max_{a' \in \mathbb{R}^{d_a}} [(A_h s + B_h a)^\top, a'^\top] G_{h+1} \begin{bmatrix} A_h s + B_h a \\ a' \end{bmatrix} = x^\top G' x.$$

The whole equation can be written as a quadratic form as well. $\square$

**Lemma 12.** *Let $A \in \mathbb{R}^{d_1 \times d_1}$ have eigenvalues $\{\lambda_i\}_{i \in [d]}$ and $B \in \mathbb{R}^{d_2 \times d_2}$ have eigenvalues $\{\mu_i\}_{i \in [d]}$. The eigenvalues of $A \otimes B$ are $\{\lambda_i \mu_j\}_{i \in [d_1], j \in d_2}$.*

## G RELAXING VISIBILITY ASSUMPTION

### G.1 TABULAR CASE

A simple but interesting case to study is the tabular case, where the value function class is the class of any bounded functions, i.e. $\mathcal{F}_h = \{f : \mathcal{S} \times \mathcal{A} \mapsto [0,1]\}$. A commonly studied family of multitask RL is the MDPs that share the same transition probability, while they have different reward functions, this problem is studied in a related literature called reward-free exploration (Jin et al., 2020a; Wang et al., 2020; Chen et al., 2022a). Specifically, (Jin et al., 2020a) propose to learn $S \times H$ sparse reward MDPs separately and generates an offline dataset, with which one can learn a near-optimal policy for any potential reward function. With a similar flavor, we show that any superset of the $S \times H$ sparse reward MDPs has low myopic exploration gap. Though the tabular case is a special case of the linear MDP case, the lower bound we derive for the tabular case is slightly different, which we show in the following section.

We first give a formal definition on the sparse reward MDP.

**Definition 9** (Sparse Reward MDPs). *Let $\mathcal{M}$ be a set of MDPs sharing the same transition probabilities. We say $\mathcal{M}$ contains all the sparse reward MDPs if for each $s, h \in \mathcal{S} \times [H]$, there exists some MDP $M_{s,h} \in \mathcal{M}$, such that $R_{h',M_{s,h}}(s',a') = \mathbb{1}(s = s', h = h')$ for all $s', a', h'$.*

To show a lower bound on the myopic exploration gap, we make a further assumption on the occupancy measure $\mu_h^\pi(s,a) := \Pr_\pi(s_h = s, a_h = a)$, the probability of visiting $s, a$ at the step $h$ by running policy $\pi$.

**Assumption 5** (Lower bound on the largest achievable occupancy measure). *For all $s, h \in \mathcal{S} \times [H]$, we assume that $\max_\pi \mu_h^\pi(s) \geq b_1$ for some constant $b$ or $\max_\pi \mu_h^\pi(s) = 0$.*

Assumption 5 guarantees that any $\beta$-optimal policy (with $\beta < b_1$) is not a vacuous policy and it provides a lower bound on the corresponding occupancy measure. We will discuss later in Appendix G on how to remove this assumption with an extra $S \times H$ factor on the sample complexity bound.

**Proposition 5.** *Consider a set of sparse reward MDP as in Definition 9. Assume Assumption 5 is true. For any $\beta \leq b_1/2$ and $f \in \mathcal{F}_\beta$, we have $\alpha(f, \mathcal{F}, \mathcal{M}) \geq \bar{\alpha}$ for some constant $\bar{\alpha} = \sqrt{\beta^2/(2e|\mathcal{M}|AH)}$ by choosing $\epsilon_h = 1/h$.*

*Proof.* We prove this lemma in a layered manner. Let $h'$ be the minimum step such that there exists some $M_{s,h'}$ is $\beta$-suboptimal. By definition, in the layer $h' - 1$, all the MDPs are $\beta$-suboptimal, in which case $\boldsymbol{\pi}_{M_{s,h'-1}}$ visits $(s, h'-1)$ with a probability at least $b/2$. Now we show that the optimal policy $\pi^*_{M_{s,h'}}$ of a suboptimal MDP $M_{s,h'}$ has lower bounded occupancy ratio.

For a more concise notation, we let $M' = M_{s,h'}$. Note that

$$\mu_{h'}^{\pi^*_{M'}}(s) = \sum_{s' \in \mathcal{S}} \mu_{h'-1}^{\pi^*_{M'}}(s') P_{h'-1}(s \mid s', \pi^*_{M'}(s'))$$

$$\leq \sum_{s' \in \mathcal{S}} \max_{\pi \in \Pi} \mu_{h'-1}^{\pi}(s') P_{h'-1}(s \mid s', \pi^*_{M'}(s'))$$

(By the fact that $\mu_{h'-1}^{\boldsymbol{\pi}_{M_{s',h'-1}}}(s')$ is $\beta$-optimal policy of $M_{s',h'-1}$)

$$\leq \sum_{s' \in \mathcal{S}} \frac{b_1}{b_1 - \beta} \mu_{h'-1}^{\boldsymbol{\pi}_{M_{s',h'-1}}}(s') P_{h'-1}(s \mid s', \pi^*_{M'}(s'))$$

$$\leq \sum_{s' \in \mathcal{S}} \frac{b_1|\mathcal{M}|A}{(b_1 - \beta)(1 - \epsilon)^{h'-1}\epsilon} \mu_{h'-1}^{\text{expl}(\boldsymbol{\pi})}(s') P_{h'-1}(s \mid s', \text{expl}(\boldsymbol{\pi})(s'))$$

$$= \frac{b_1|\mathcal{M}|A}{(b_1 - \beta)(1 - \epsilon)^{h'-1}\epsilon} \mu_{h'}^{\text{expl}(\boldsymbol{\pi})}(s)$$

The occupancy measure ratio can be upper bounded by $c = \frac{b_1|\mathcal{M}|A}{(b_1-\beta)(1-\epsilon)^{h'-1}\epsilon}$. Then the myopic exploration gap can be lower bounded by

$$\frac{\beta}{\sqrt{c}} = \sqrt{\frac{(b_1 - \beta)\beta^2(1 - \epsilon)^{h'-1}\epsilon}{b_1|\mathcal{M}|A}} \geq \sqrt{\frac{\beta^2(1 - \epsilon)^{h'-1}\epsilon}{2|\mathcal{M}|A}}.$$

To proceed, we choose $\epsilon_h = 1/h$, which leads to $(1 - \epsilon_h)^{h-1}\epsilon \geq 1/(eH)$. $\qquad\square$

Plugging this into Theorem 1, we achieve a sample complexity bound of $\mathcal{O}(S^2 A H^5/\beta^2)$, with $|\mathcal{M}| = SH$. This is not a near-optimal bound for reward-free exploration (a fair comparison in our setup). This is because the sample complexity bound in Theorem 1 is not tailored for tabular case.

## G.2 REMOVING COVERAGE ASSUMPTION

Though Assumption 2 and Assumption 4 are relatively common in the literature, we have not seen an any like Assumption 5. In fact, Assumption 5 is not a necessary condition for sample-efficient myopic exploration as we will discuss in this section. The main technical invention is to construct a mirror transition probability that satisfies the conditions in Assumption 5. However, we will see that a inevitable price of an extra $SH$ factor has to be paid.

To illustrate the obstacle of removing Assumption 5, recall that the proof of Proposition 5 relies on the fact that all $\beta$-optimal policies guarantee a non-zero probability of visiting the state corresponding to their sparse reward with $\beta < b_1/2$. Without Assumption 5, a $\beta$-optimal policy can be an arbitrary policy. At the step $h$, we have at most $S$ such MDPs, which may accumulate an irreducible error of $S\beta$, which means that at the round $h + 1$, we can only guarantee $S\beta$-optimal policies. An naive adaptation will require us to set the accuracy $\beta' = \beta/S^H$ in order to guarantee a $\beta$ error in the last step. The following discussion reveals that the error does not accumulate in a multiplicative way.

**Mirror MDP construction.** It is helpful to consider a mirror transition probability modified from our original transition probability. We denote the original transition probability by $P = \{P_h\}_{h \in [H]}$. Consider a new MDP with transition $P' = \{P'_h\}_{h \in [H]}$ and state space $\mathcal{S}' = \mathcal{S} \cup \{s_0\}$, where $s_0$ is a dummy state. We initialize $P'$ such that

$$P'_h(s' \mid s, a) = P_h(s' \mid s, a) \text{ for all } s', s, a, h, \text{ where } s', s \neq s_0, \text{ and } P'_h(s_0 \mid s_0, \cdot) = 1 \qquad (7)$$

Starting from $h = 1$, we update $P'_h$ by a forward induction according to Algorithm 2. The design principle is to direct the probability mass of visiting $(s, h + 1)$ to $(s_0, h + 1)$, whenever the maximal probability of visiting $(s, h + 1)$ is less than $\beta$.

---

**Algorithm 2** Creating Mirror Transitions

**Input:** Original Transition $P$, threshold $\beta > 0$.
Initialize $P'$ according to (7)
**for** $h = 1, 2, \ldots, H - 1$ **do**
    **for** each $s \in \mathcal{S}$ such that $\max_\pi \mu'^\pi_{h+1}(s) \leq \beta$ **do**
        $P'_h(s_0 \mid \tilde{s}, \tilde{a}) \leftarrow P'_h(s_0 \mid \tilde{s}, \tilde{a}) + P'_h(s \mid \tilde{s}, \tilde{a})$ for each $\tilde{s}, \tilde{a}$.
        $P'_h(s \mid \tilde{s}, \tilde{a}) \leftarrow 0$ for each $\tilde{s}, \tilde{a}$.
    **end for**
**end for**
**Return** $P'$

---

By definition of $P'$, we have two nice properties.

**Proposition 6.** *For any $h \in [H]$, $s \in \mathcal{S}$, we have $\max_\pi \mu'^\pi_h(s) = 0$ or $\max_\pi \mu'^\pi_h(s) > \beta$.*

Thus, $P'$ nicely satisfies our Assumption 5. We also have that $P'$ is not significantly different from $P$.

**Proposition 7.** *For any policy $\pi$, $\mu'^\pi_h(s) \geq \mu^\pi_h(s) - HS\beta$. Further more, any $(SH + 1)\beta$-suboptimal policy for $P$ is at least $\beta$-suboptimal for $P'$ with respect to the same reward.*

*Proof.* We simply observe that $\max_\pi \mu'^\pi_h(s_0) \leq (h - 1)S\beta$. This is true since at any round, we have at most $S$ states with $\max_\pi \mu'^\pi(s) \leq \beta$, all the probability that goes to $s$ will be deviated to $s_0$. Therefore, for any $\pi$

$$\mu'^\pi_{h+1}(s_0) \leq \mu'^\pi_h(s_0) + S\beta.$$

$\qquad\square$

Therefore, any $(SH+1)\beta$-suboptimal policy for $P$ has the myopic exploration gap of $\beta$-suboptimal policy for $P'$.

**Theorem 4.** *Consider a set of sparse reward MDP as in Definition 9. For any $\beta \in (0,1)$ and $f \in \mathcal{F}_\beta$, we have $\alpha(f, \mathcal{F}, \mathcal{M}) \geq \bar\alpha$ for some constant $\bar\alpha = \Omega(\sqrt{\beta^2/(|\mathcal{M}|AS^2H^3)})$ by choosing $\epsilon_h = 1/(h+1)$.*

## H  CONNECTIONS TO DIVERSITY

Diversity has been an important consideration for the generalization performance of multitask learning. How to construct a diverse set, with which we can learn a model that generalizes to unseen task is studied in the literature of multitask supervised learning.

Previous works (Tripuraneni et al., 2020; Xu & Tewari, 2021) have studied the importance of diversity in multitask representation learning. They assume that the response variable is generated through mean function $f_t \circ h$, where $h$ is the representation function shared by different tasks and $f_t$ is the task-specific prediction function of a task indexed by $t$. They showed that diverse tasks can learn the shared representation that generalizes to unseen downstream tasks. More specifically, if $f_t \in \mathcal{F}$ is a discrete set, a diverse set needs to include all possible elements in $\mathcal{F}$. If $\mathcal{F}$ is the set of all bounded linear functions, we need $d$ tasks to achieve a diverse set. Note that these results align with the results presented in the previous section. *Could there be any connection between the diversity in multitask representation learning and the efficient myopic exploration?*

Xu & Tewari (2021) showed that Eluder dimension is a measure for the hardness of being diverse. Here we introduce a generalized version called distributional Eluder dimension (Jin et al., 2021a).

**Definition 10** ($\varepsilon$-independence between distributions). *Let $\mathcal{G}$ be a class of functions defined on a space $\mathcal{X}$, and $\nu, \mu_1, \ldots, \mu_n$ be probability measures over $\mathcal{X}$. We say $\nu$ is $\varepsilon$-independent of $\{\mu_1, \mu_2, \ldots, \mu_n\}$ with respect to $\mathcal{G}$ if there exists $g \in \mathcal{G}$ such that $\sqrt{\sum_{i=1}^n (\mathbb{E}_{\mu_i}[g])^2} \leq \varepsilon$, but $|\mathbb{E}_\nu[g]| > \varepsilon$*

**Definition 11** (Distributional Eluder (DE) dimension). *Let $\mathcal{G}$ be a function class defined on $\mathcal{X}$, and $\Pi$ be a family of probability measures over $\mathcal{X}$. The distributional Eluder dimension $\dim_{\mathrm{DE}}(\mathcal{G}, \Pi, \varepsilon)$ is the length of the longest sequence $\{\rho_1, \ldots, \rho_n\} \subset \Pi$ such that there exists $\varepsilon' \geq \varepsilon$ where $\rho_i$ is $\varepsilon'$-independent of $\{\rho_1, \ldots, \rho_{i-1}\}$ for all $i \in [n]$.*

**Definition 12** (Bellman Eluder (BE) dimension (Jin et al., 2021)). *Let $\mathcal{E}_h\mathcal{F}$ be the set of Bellman residuals induced by $\mathcal{F}$ at step $h$, and $\Pi = \{\Pi_h\}_{h=1}^H$ be a collection of $H$ probability measure families over $\mathcal{X} \times \mathcal{A}$. The $\varepsilon$-Bellman Eluder dimension of $\mathcal{F}$ with respect to $\Pi$ is defined as*

$$\dim_{\mathrm{BE}}(\mathcal{F}, \Pi, \varepsilon) := \max_{h \in [H]} \dim_{\mathrm{DE}}(\mathcal{E}_h\mathcal{F}, \Pi, \varepsilon)$$

**Constructing a diverse set.** For each $h \in [H]$, consider a sequence of functions $f_1, \ldots, f_d \in \mathcal{F}$, such that the induced policy $(\pi^{f_i})_{i \in [d]}$ generates probability measures $(\mu_{h+1}^{f_i})_{i \in [d]}$ at the step $h+1$. Let $(\mu_{h+1}^{f_i})_{i \in [d]}$ be $\epsilon$-independence w.r.t the function class $\mathcal{E}_h\mathcal{F}$ between their predecessors. By the definition of BE dimension, we can only find at most $\dim_{\mathrm{DE}}(\mathcal{E}_h\mathcal{F}, \Pi, \varepsilon)$ of these functions. Then conditions in Definition 3 is satisfied with $c = 1/(dH)$.

**Revisiting linear MDPs.** The task set construction in 7 seems to be quite restricted as we require a set of standard basis. One might conjecture that a task set $M_{i,h}$ with full rank $[\theta_{1,h}, \ldots, \theta_{d,h}]$ will suffice. From what we discussed in the general case, we will need the occupancy measure generated by the optimal policies for these MDPs to be $\epsilon$-independent and any other distribution is $\epsilon$-dependent. This is generally not true even if the reward parameters are full rank. To see this, let us consider a tabular MDP case with two states $\{1, 2\}$, where at the step $h$, we have two tasks $M_1$, $M_2$, with $R_{h,M_1}(s,a) = \mathbb{1}[s=1]$ and $R_{h,M_2}(s,a) = 0.51\mathbb{1}[s=1] + 0.49\mathbb{1}[s=2]$. This gives $\theta_{h,M_1} = [1, 0]$ and $\theta_{h,M_2} = [0.49, 0.51]$ as shown in Figure 3.

Construct the MDP such that the transition probability and action space any policy either visit state 1 or state 2 at the step $h$. Then the optimal policies for both tasks are the same policy which visits state 1 with probability one, even if the reward parameters $[\theta_{h,M_1}, \theta_{h,M_2}]$ are full-rank.

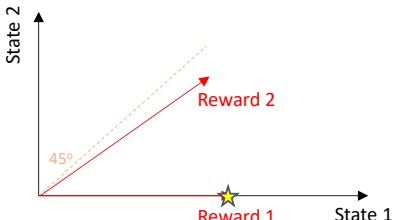

Figure 3: An illustration of why a full-rank set of reward parameters does not achieve diversity. The red arrows are two reward parameters and the star marks the generated state distributions of the optimal policies corresponding to the two rewards at the step $h$. Since both optimal policies only visit state 1, they may not provide a sufficient exploration for the next time step $h + 1$.

# I    IMPLICATIONS OF DIVERSITY ON ROBOTIC CONTROL ENVIRONMENTS

In this section, we conduct simulation studies on robotic control environments with practical interests. Since myopic exploration has been shown empirically efficient in many problems of interest, we focus on the other main topic–diversity. We investigate how our theory guides a diverse task set selection. More specifically, our prior analysis on Linear MDPs suggests that a diverse task set should prioritize tasks with full-rank feature covariance matrices. We ask whether tasks with a more spread spectrum of the feature covariance matrix lead to a better training task set. *Note that the goal of this experiment is not to show the practical interests of Algorithm 1. Instead, we are revealing interesting implications of the highly conceptual definition of diversity in problems with practical interests.*

**Environment and training setup.** We adopt the BipedalWalker environment from (Portelas et al., 2020). The learning agent is embodied into a bipedal walker whose motors are controllable with torque (i.e. continuous action space). The observation space consists of laser scan, head position, and joint positions. The objective of the agent is to move forward as far as possible, while crossing stumps with varying heights at regular intervals (see Figure 4 (a)). The agent receives positive rewards for moving forward and negative rewards for torque usage. An environment or task, denoted as $M_{p,q}$, is controlled by a parameter vector $(p, q)$, where $p$ and $q$ denote the heights of the stumps and the spacings between the stumps, respectively. Intuitively, an environment with higher and denser stumps is more challenging to solve. We set the parameter ranges for $p$ and $q$ as $p \in [0, 3]$ and $q \in [0, 6]$ in this study.

The agent is trained by Proximal Policy Optimization (PPO) (Schulman et al., 2017) with a standard actor-critic framework (Konda & Tsitsiklis, 1999) and with Boltzmann exploration that regularizes entropy. Note that Boltzmann exploration strategy is another example of myopic exploration, which is commonly used for continuous action space. Though it deviates from the $\epsilon$-greedy strategy discussed in the theoretical framework, we remark that the theoretical guarantee outlined in this paper can be trivially extend to Boltzmann exploration. The architecture for the actor and critic feature extractors consists of two layers with 400 and 300 neurons, respectively, and Tanh (Rumelhart et al., 1986) as the activation function. Fully-connected layers are then used to compute the action and value. We keep the hyper-parameters for training the agent the same as Romac et al. (2021).

## I.1    INVESTIGATING FEATURE COVARIANCE MATRIX

We denote by $\phi(s, a)$ the output of the feature extractor. We evaluate the extracted feature at the end of the training generated by near-optimal policies, denoted as $\pi$, on 100 tasks with different parameter vectors $(p, q)$. We then compute the covariance matrix of the features for each task, denoted as $V_{p,q} = \mathbb{E}_\pi^{M_{p,q}} \sum_{h=1}^H \phi(s_h, a_h)\phi(s_h, a_h)^T$, whose spectrums are shown in Figure 4 (b) and (c).

By ignoring the extremely large and small eigenvalues on two ends, we focus on the largest 100-200 dimension, where we observe that the height of the stumps $p$ has a larger impact on the distribution of eigenvalues. In Figure 4 (b), we show the boxplot of the log-scaled eigenvalues of 100-200 dimensions for environments with different heights, where we average spacings. We observe that the eigenvalues can be 10 times higher for environments with an appropriate height (1.0-2.3), compared

to extremely high and low heights. However, the scales of eigenvalues are roughly the same if we control the spacings and take average over different heights as shown in Figure 4 (c). This indicates that choosing an appropriate height is the key to properly scheduling tasks.

In fact, the task selection coincidences with the tasks selected by the state-of-the-art Automatic Curriculum Learning (ACL). We investigate the curricula generated by ALP-GMM (Portelas et al., 2020), a well-established curriculum learning algorithm, for training an agent in the BipedalWalker environment for 20 million timesteps. Figure 4 (d) gives the density plots of the ACL task sampler during the training process, which shows a significant preference over heights in the middle range, with little preference over spacing.

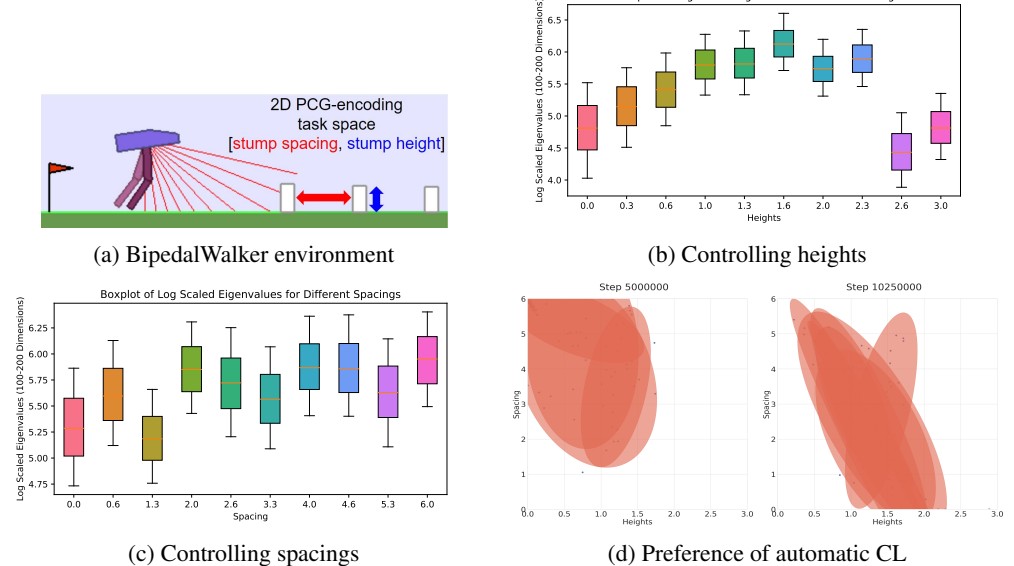

Figure 4: **(a)** BipedalWalker Environment with different stump spacing and heights. **(b-c)** Boxplots of the log-scaled eigenvalues of sample covariance matrices of the trained embeddings generated by the near optimal policies for different environments. In (b), we take average over environments with the same height while in (c), over the same spacing. **(d)** Task preference of automatically generated curriculum at 5M and 10M training steps respectively. The red regions are the regions where a task has a higher probability to be sampled.

**Training on different parameters.** To further validate our finding, we train the same PPO agent with different means of the stump heights and see that how many tasks does the current agent master. As we argued in the theory, a diverse set of tasks provides good behavior policies for other tasks of interest. Therefore, we also test how many tasks it could further master if one use the current policy as behavior policy for fine-tuning on all tasks. The number of tasks the agent can master by learning on environments with heights ranging in [0.0, 0.3], [1.3, 1.6], [2.6, 3.0] are 28.1, 41.6, 11.5, respectively leading to a significant outperforming for diverse tasks ranging in [1.3, 1.6]. Table 1 gives a complete summary of the results.

Table 1: Training on different environment parameters. Each row represents a training scenario, where the first two columns are the range of sampled parameters. The mastered tasks are out of 121 evaluated tasks with the standard deviation calculated from ten independent runs.

| Obstacle spacing | Stump height | Mastered task |
|---|---|---|
| [2, 4] | [0.0, 0.3] | $28.1 \pm 6.1$ |
| [2, 4] | [1.3, 1.6] | $41.6 \pm 9.8$ |
| [2, 4] | [2.6, 3.0] | $11.5 \pm 10.9$ |

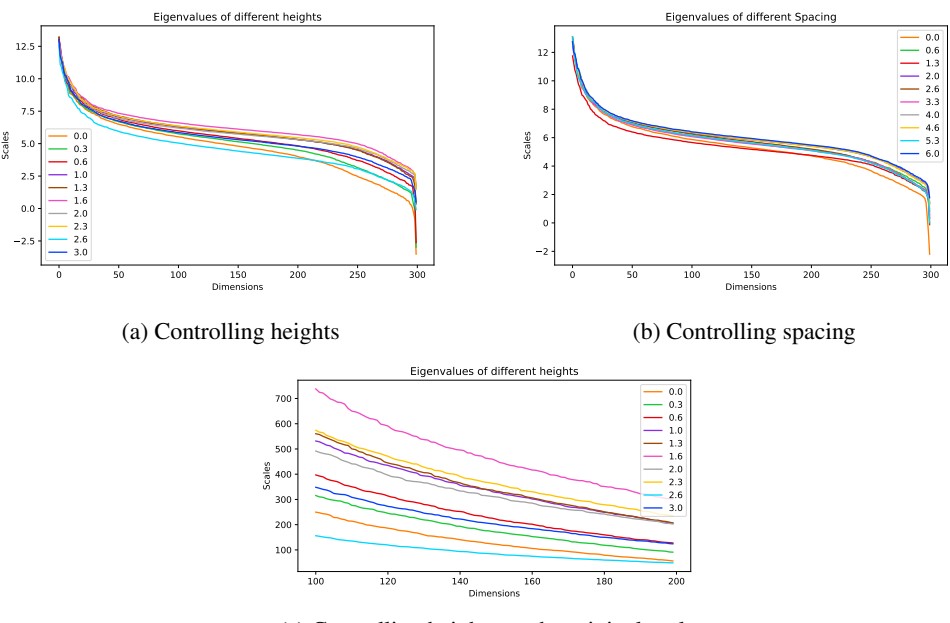

(a) Controlling heights

(b) Controlling spacing

(c) Controlling heights on the original scale

Figure 5: **(b-c)** Log-scaled eigenvalues of sample covariance matrices of the trained embeddings generated by the near optimal policies for different environments.