# OpenReview forum: "Sample Efficient Myopic Exploration Through Multitask Reinforcement Learning with Diverse Tasks"
_ICLR.cc/2024/Conference — ICLR 2024 poster_

### Official Review · Reviewer_GSze · 2023-10-31

**Soundness:** 3 good
**Presentation:** 2 fair
**Contribution:** 3 good
**Rating:** 6
**Confidence:** 3

**Summary:**

This paper studies the use of multitask training to enable myopic (overly simplistic) exploration methods to discover solutions to difficult MDPs which would otherwise not be tractable to learn directly.

**Strengths:**

The core argument of this paper is salient and interesting, and seems to be well supported by the theoretical arguments. The experimental validation in a deep RL context is also appreciated in what is otherwise a theory paper.

**Weaknesses:**

I'm not well versed in recent theoretical/tabular RL exploration literature, so I can't speak very well to the novelty and significance there, but in a deep RL context this work seems relevant, but also very closely related to existing approaches such as goal-conditioned RL (Hindsight Experience Replay in particular) and to some extent automatic curriculum generation methods.

I think these connections are very interesting, and this theoretical analysis isn't redundant with that work, but it does leave me feeling like this paper would me more interesting/have a stronger contribution if that connection were explored more. As it is I'm left feeling that while this specific argument is to my knowledge novel, it overlaps a lot with prior work.

In addition, I also felt like the presentation of the paper needs some significant polish. There's some issues with grammar and odd phrasing throughout the paper, and I while I appreciate the intuition provided for various definitions/theorems I felt like I frequently lost the thread on those.

Overall, I think this is solid work that could be high impact, but it needs a little more polish to really shine. As such I'm inclined to recommend rejection, but I also admit that I don't have a good sense of the impact on the tabular RL exploration literature, so I will caveat that I can't properly evaluate that aspect and I will defer to other reviewers there.

**Questions:**

-While I generally follow the argument, this paper has some rough grammar and odd word choice in places. I'd recommend a thorough editing pass to improve the language.

-The explanation of the multitask setup in Section 2.1 confused me as to how the tasks are getting selected. Is there a structure or order in which tasks are chosen among M?

-Likewise Definition 1 is a little confusing. C is a function of beta and delta? I'm confused as to why sample complexity doesn't depend on either the algorithm itself or the task(s) being learned. The following paragraph seems to think C is a function of the MDP, but this isn't part of the definition.

-How does algorithm 1 differ from the cited Zhumabekov 2023 policy ensemble method? It seems like algorithm 1 is essentially an ensemble of policies, of which one is sampled for each episode?

-How does this idea of multitask myopic exploration differ from normal goal-directed RL methods like hindsight experience replay? The motivating example in Figure 1 seems roughly in line with such methods, and seems like it should share their limitations (e.g. large state spaces and low-dimensional manifolds of interesting/human-relevant tasks).

 -I find the term "myopic exploration gap" a little confusing. I understand the intuition (how much could a myopic exploration method improve upon the current best policy at any given time), but I wouldn't call that a gap. Maybe something like "myopic exploration potential?" A gap would imply it is comparing myopic exploration to another (optimal?) exploration algorithm. I know this term is coming from previous literature, but it seems confusing unless I'm misunderstanding the definition here.

-Doesn't PPO (like all on-policy policy gradient methods) have issues with epsilon greedy optimization due to it's off-policyness? How did you resolve this issue for the experiments?

-In the tabular case, multitask myopic exploration relies on coverage assumptions in the space of possible tasks (if I understand correctly), but it's not tractable to assume this in the deep RL case. Did this factor come up in the BipedalWalker experiments at all? Do the assumptions (mostly, at least) hold?

-Some more details/analysis on the deep RL experiments in the main paper would be appreciated, such as performance/training curves. I realize the focus of the paper is tabular/theoretical, but this topic has a lot of connections to methods used in deep RL (such as goal-directed RL and automatic curricula, as noted), and in my opinion exploring that connection further would be very interesting.

---

> ### Author Response · Authors · 2023-11-12
>
> We thank the reviewer for their valuable suggestions from a deep RL perspective. Here are our detailed point-by-point responses.
>
> **Responses to the General Comment in Weakness:**
>
> We value the reviewer’s contextualization within the realm of deep RL. Indeed, multi-goal RL serves a motivating example of our proposed theoretical framework. HER, as proven to improve sample complexity, implicitly tackles a multitask RL problem, as articulated in Section 3. Specifically, Algorithm 1 encapsulates a form of HER within a multi-goal RL setting.
>
> **Connection to HER and multi-goal RL:** In a multi-goal RL setting, all tasks share identical state space, action space, and transition function, with the only difference residing in the reward function. Algorithm 1 exemplifies this by sampling a trajectory on task $M$ through a mixture of all greedy policies from other tasks. Notably, this exploration strategy is akin to randomly selecting task $M'$ and collecting a trajectory on $M'$ using its own epsilon-greedy policy, followed by relabeling rewards to simulate a trajectory on $M$.
>
> **Connection to curriculum learning:** Although Algorithm 1 does not explicitly implement curriculum learning by assigning preferences to tasks, an improvement could be made through adaptive task selection. If an algorithm with adaptive task selection achieves a sample complexity $\mathcal{C}$, Algorithm 1 should have a sample complexity of at most $\mathcal{C} |\mathcal{M}|^2$. Intuitively, if a curriculum selects tasks through an order of $M_1, \dots, M_T$, then at the each round, the algorithm uses the epsilon-greedy policy of MDP $M_{t-1}$ to explore $M_t$. This exploration is implicitly included in Algorithm 1 because Algorithm 1 explores all the MDPs in each round with the mixture of all epsilon-greedy policies.
>
> This means that the sample-complexity of Algorithm 1 provides an upper bound on the sample complexity of underlying optimal task selection and in this way our theory provides some insights on the success of the curriculum learning. Designing a theoretically grounded algorithm for near-optimal adaptive task selection is a non-trivial improvement, and we defer this discussion to future work.
>
> **Responses to Questions:**
>
> **Question 1: While I generally follow the argument, this paper has some rough grammar and odd word choices in places. I'd recommend a thorough editing pass to improve the language.**
>
> Response: We appreciate the observation regarding language issues and commit to a comprehensive editing pass to enhance the final version's clarity and coherence.
>
> **Question 2: The explanation of the multitask setup in Section 2.1 confused me as to how the tasks are getting selected. Is there a structure or order in which tasks are chosen among M?**
>
> Response: In the general multitask RL setup in Section 2.1, no assumptions are made regarding how tasks are selected. Algorithms have the flexibility to choose any task for trajectory collection in each round. To simplify the task selection in Algorithm 1, we explore all tasks simultaneously (line 3), avoiding the need for an adaptive task selection.
>
> **Question 3: Likewise, Definition 1 is a little confusing. C is a function of beta and delta? I'm confused as to why sample complexity doesn't depend on either the algorithm itself or the task(s) being learned.**
>
> Response: We appreciate the clarification question. The sample complexity $C$ is indeed a function of $\beta$, $\delta$, the algorithm, and the environment $\mathcal{M}$. While we did not explicitly state the dependence on the algorithm and environment to maintain simplicity, we acknowledge the importance of this clarification.
>
> **Question 4: How does algorithm 1 differ from the cited Zhumabekov 2023 policy ensemble method?**
>
> Response: While both Algorithm 1 and Zhumabekov 2023 share a similar high-level idea of ensembling policies, their implementation differs. In Algorithm 1, policies are mixed trajectory-wise, with one policy randomly selected for the entire episode. In contrast, Zhumabekov 2023 mixes policies on a step-wise basis for a continuous action space, selecting actions as a weighted average of actions chosen by different policies. We conjecture that this distinction impacts the worst-case sample complexity, with Algorithm 1 enjoying the guarantees outlined.
>
> **Question 5: How does this idea of multitask myopic exploration differ from normal goal-directed RL methods like hindsight experience replay?**
>
> Response: As elucidated in the Weakness section, Algorithm 1 implicitly incorporates HER in a multi-goal setting. It converges with methods like HER, sharing a common approach to myopic exploration.

---

> > ### Author Response · Authors · 2023-11-12
> >
> > **Question 6: I find the term "myopic exploration gap" a little confusing. Maybe something like "myopic exploration potential?"**
> >
> > Response: We agree with the suggested term "myopic exploration potential" and recognize its potential to more accurately capture the underlying intuition. Since "myopic exploration gap", we would like to keep the current name to avoid confusion for readers who know the previous literature.
> >
> > **Question 7: Doesn't PPO have issues with epsilon greedy optimization due to its off-policyness? How did you resolve this issue for the experiments?**
> >
> > Response: If by off-policyness, you refer to PPO running $N$ parallel actors, it shouldn't significantly disrupt exploration when $N \ll T$ with $T$ being the total number of runs.
> >
> > **Question 8: In the tabular case, multitask myopic exploration relies on coverage assumptions. Did this factor come up in the BipedalWalker experiments? Do the assumptions hold?**
> >
> > Response: We think that insights from linear MDPs may be relevant in deep RL, where a shared representation function is used for all tasks, an analog to the representation learning setting in deep RL. Our exploration of the embedding covariance matrix spectrum in Figure 2 aligns with findings from automatic curriculum learning task selection. Although not an exact translation because the real environment is not a linear MDP, the theoretical insights from linear MDPs offer valuable considerations for the deep RL setting.
> >
> > **Question 9: Some more details/analysis on the deep RL experiments in the main paper would be appreciated.**
> >
> > Response: We appreciate the suggestion and commit to including more comprehensive details, including training curves and performance metrics, in the revised version. The revised manuscript will offer a deeper exploration of connections to methods used in deep RL, enhancing the overall depth of the paper.
> >
> > **General Response:** We acknowledge the need for more explicit discussions on connections to HER and automatic curriculum learning and commit to including these in the final version. We believe that the detailed responses provided here contribute to clarifying these connections.

---

> ### Comment · Reviewer_GSze · 2023-11-21
> **Response to Rebuttal**
>
> Thanks for your comments and clarifications! Since most of my issues were with the presentation or drawing connections to related algorithms (which it sounds like you were aware of already, pleasantly), I think a revised manuscript with additions/changes along the lines you proposed would raise my score, if one can be assembled in time (and if not I encourage you to revise and resubmit the paper to another conference).
>
> Re: question 7, I was thinking of how in standard (e.g. off-policy DQN style) epsilon-greedy a random action is chosen with probability epsilon at every timestep, which in this case results in a 1/epsilon fraction of actions PPO is trained on being off-policy actions (since they were chosen at random). In theory as well as my experience (inadvertently implementing epsilon-greedy exploration in the context of PPO-like algorithms), this causes unstable gradients and policy distribution/performance collapse, particularly late in training when the policy is relatively converged and randomly-selected actions often have an extremely small probability under the policy. Was this an issue for your PPO experiments, or am I misunderstanding what is meant by epsilon-greedy exploration in that experiment?

---

> > ### Author Response · Authors · 2023-11-21
> >
> > Dear reviewer,
> >
> > Thank you for your additional comments. We appreciate your thorough review.  We have made revisions to the manuscript to address your concerns regarding the literature on multi-goal RL and curriculum learning. These modifications are clearly indicated in red within the revised document, and we believe they significantly enhance the presentation and discussion of related algorithms.
> >
> > Regarding your query (question 7), we sincerely apologize for the mistake in stating that our PPO implementation uses $\epsilon$-greedy exploration (line 326).  In fact, we used a standard PPO implementation with entropy regularization as exploration. Such an exploration strategy is also myopic, but deviates from the major $\epsilon$-greedy strategy discussed in the theoretical framework. We remark that the theoretical guarantees outlined in this paper can be trivially extended to Algorithm 1 with entropy regularization exploration.
> >
> > We want to reiterate that the goal of the experiments (line 313-315) is not to demonstrate the practical interests of Algorithm 1. Instead, we are revealing interesting implications of the highly conceptual definition of diversity in problems with practical interests.
> >
> > To clarify the distinctions between the experimental setting and theoretical setups, we provide the following comparison:
> >
> > |                      | Experiment             | Theory            |
> > | -------------------- | ---------------------- | ----------------- |
> > | Exploration strategy | Entropy regularization | $\epsilon$-greedy |
> > | MDP                  | Bipedal Walker            | Linear with non-linear embedding function           |
> >
> > We hope these clarifications address your concerns. Thank you for your continued engagement with our work.

---

> > > ### Comment · Reviewer_GSze · 2023-11-22
> > > **Response to Revised Manuscript**
> > >
> > > Thanks for the further response! Using an entropy regularization bonus makes sense for PPO, yeah. Intuitively I agree the exact form of the myopic exploration algorithm shouldn't affect the conclusions (and as an aside I wonder if there's future work that could be done to define similar bounds for less myopic exploration strategies?), just wanted to make sure there wasn't something odd going on with the PPO experiments.
> > >
> > > Thanks for making the connections to HER/curriculum learning approaches more explicit! I will quibble that the current placement of those paragraphs feels a little out of place (maybe the connection could be discussed initially in the introduction, to establish the significance of this work for those topics, then revisited briefly in Section 3?), and there's more in-depth connections and experiments that could be developed, but I'm happy to call this sufficient for one paper, and encourage you to build upon these results further in the future. As such, I've raised my score to recommend acceptance (with encouragement to integrate the connections to those deep RL topics better for the camera-ready draft).

---

> > > > ### Author Response · Authors · 2023-11-22
> > > >
> > > > We thank reviewer for raising the score. We could not do an in-depth discussion on the connections to HER/Curriculum Learning due to the page limit in the rebuttal phase. We will improve this in the camera-ready version when we have an extra page. We will also discuss the future work that makes a more concrete connection between our theory and the real scenarios in Deep RL.

---

### Official Review · Reviewer_ZHzf · 2023-11-01

**Soundness:** 3 good
**Presentation:** 3 good
**Contribution:** 3 good
**Rating:** 6
**Confidence:** 4

**Summary:**

This paper studies the statistical efficiency of exploration in Multitask Reinforcement Learning (MTRL). The authors show that when an agent is trained on a sufficiently diverse set of tasks, a generic policy-sharing algorithm with myopic exploration design like ϵ-greedy that are inefficient in general can be sample-efficient for MTRL. To validate the role of diversity, the authors conduct experiments on synthetic robotic control environments, where the diverse task set aligns with the task selection by automatic curriculum learning, which is empirically shown to improve sample-efficiency.

**Strengths:**

1.	The paper shows that when an agent is trained on a sufficiently diverse set of tasks, a generic policy-sharing algorithm with myopic exploration design like ϵ-greedy that are inefficient in general can be sample-efficient for MTRL.
2.	The paper is well-written and easy to follow.
3.	To the best of my knowledge, this is the first theoretical demonstration of the "exploration benefits" of MTRL, which is insightful for future research on efficient exploration in deep RL.

**Weaknesses:**

1.	The assumption that the task set is adequately diverse may be too strong in deep RL. Although the authors discuss implications of diversity in deep RL, it remains unclear to me. The authors may want to provide more insight into how to define and design a diverse task set for efficient exploration in deep RL.

**Questions:**

Please refer to Weaknesses for my questions.

---

> ### Author Response · Authors · 2023-11-12
>
> We thank the reviewer for their thoughtful examination of our paper. We acknowledge the need for more comprehensive discussions regarding the concept of diversity in deep RL.
>
> Towards this direction, our simulation study provides a potential connection between diversity in the linear MDP case and the diversity in deep RL. In short, in linear MDP cases, we require that the optimal policies for different tasks induce a full-rank feature covariance matrix at the each step $h$. Our simulation study finds a benefit of having a more spread spectrum of feature covariance matrix in bipedal walker environment (medium range of stump heights) illuminating a potential way of investigating the diversity of tasks in deep RL.
>
> To attain diversity in deep RL, one may apply the automatic curriculum learning (ACL) algorithm, which has been shown to coincidence with our notion of diversity in bipedal walker environment. It appears that ACL may implicitly optimize the diversity notion proposed in our paper. As part of our future work, we plan to formalize this observed similarity between ACL task selection and diversity from a theoretical perspective. Moreover, our work can be viewed as providing an explanation for the success of ACL in multitask RL tasks.

---

> > ### Comment · Area_Chair_1vhQ · 2023-11-22
> > **Please respond to the author reply**
> >
> > Dear reviewer, please do respond to the author reply and let them know if this has answered your questions/concerns.

---

### Official Review · Reviewer_q9zq · 2023-11-02

**Soundness:** 3 good
**Presentation:** 4 excellent
**Contribution:** 3 good
**Rating:** 8
**Confidence:** 3

**Summary:**

This paper studies the potential exploration benefits of multitask reinforcement learning from a theoretical perspective. This paper shows that when the set of tasks is diverse enough (measured by multitask MEG), a generic policy-sharing algorithm with myopic exploration is sample-efficient. Importantly, such myopic exploration is common in practice, and computationally efficient (unlike GOLF which requires solving nested optimization oracles). The paper also gives concrete examples of tabular/linear MDPs such that the diversity condition is satisfied. In the end, the paper validates the proposed theory with experiments and builds connections with curriculum learning.

**Strengths:**

- The general idea of this paper is novel and natural. Sample efficiency of myopic exploration is an important topic.

- The paper is very well-written.

- The theoretical results are sound.

- Discussion on limitations and comparison with prior works are adequate.

**Weaknesses:**

- The only weakness of this paper in my opinion is that examples where multitask MEG is bounded are too restrictive. Definition 7 is a very strong requirement, and intuitively, diverse tasks can be defined more general. Moreover, the feature coverage assumption is additional since it is not needed for learning linear MDPs with strategic exploration.

**Questions:**

- Is it possible to relax Definition 7?

- The offline learning oracle solves  $f_1,...,f_h$ simultaneously. Can you do them sequentially and have similar guarantees?

---

> ### Author Response · Authors · 2023-11-12
>
> We thank the reviewer for the careful reading and insightful comments.
>
> Response to weakness: The feature coverage assumption is an additional condition included for clarity in presentation. We acknowledge it is non-essential and discuss a relaxation of this assumption for the Tabular case. In this scenario, we construct a mirror MDP (refer to Appendix F.2), introducing an extra dependence of $\sqrt{SH}$.
>
> **Question 1: Is it possible to relax Definition 7?**
>
> Response: We concede that Definition 7 is indeed restrictive. We might conjecture that we only need the $\theta$'s for the reward parameter is rank $d$. This is in fact not sufficient. We provide an example in Appendix D Figure 3. To have a lower bounded Multitask MEG, we require, at each $h$, the optimal policies of different tasks induce a full-rank covariance matrix of the feature vectors (refer to Lemma 8).
>
> More generally, a thorough discussion of diversity for general function classes is available in Appendix G. The overarching concept involves constructing a set of tasks whose optimal policies induce occupancy measures that are $\epsilon$-dependent. The size of the largest set is quantified by the BE dimension. However, the technical challenge lies in the fact that current algorithms only guarantee finding a $\beta$-optimal solution, which doesn't necessarily ensure convergence to the induced occupancy measure of the true optimal policy. There can be many interesting future works towards this direction.
>
> **Question 2: The offline learning oracle solves simultaneously. Can you do them sequentially and have similar guarantees?**
>
> Response: By "solving them sequentially," if you mean updating the value function for one task at each round, this approach is indeed feasible. We can achieve this by selecting a task in each round and collecting an episode for that task using the mixture policy (where the greedy policy from other tasks is based on the most recent update). If tasks are chosen in a fixed order ($1, \dots, |\mathcal{M}|$), we obtain guarantees similar to those outlined in Algorithm 1. There is potential for an adaptive task selection approach, akin to curriculum learning, which we leave for further exploration in future work

---

> > ### Comment · Area_Chair_1vhQ · 2023-11-22
> > **Please respond to the author reply**
> >
> > Dear reviewer, please do respond to the author reply and let them know if this has answered your questions/concerns.

---

### Official Review · Reviewer_E4JK · 2023-11-10

**Soundness:** 3 good
**Presentation:** 3 good
**Contribution:** 2 fair
**Rating:** 5
**Confidence:** 2

**Summary:**

This paper claims that in the scenario of multitask-RL, a naive exploration strategy is enough. It formalizes their intuition in Def 3 and provide the theoretical guarantee in Theorem 1.

**Strengths:**

The proof seems to be sound.

**Weaknesses:**

1. It provides a possible explanation to explain the success of naive exploration in the case of multitask RL. However, it is hard to validate such explanation.

2. The intuition of the proof is that, if we have a base policy class with good coverage, we are able to find the optimal policy by combining the base policies with naive exploration. However, it have been well known that exploration is simple when we have good coverage. Therefore, their contribution seems not novel enough.

[1]. Xie, Tengyang, Dylan J. Foster, Yu Bai, Nan Jiang, and Sham M. Kakade. "The role of coverage in online reinforcement learning." arXiv preprint arXiv:2210.04157 (2022).

**Questions:**

1. What does Def 3 mean in linear MDP?

2. Can you provide an example where Def 3 holds and the coverage of the base policies is poor?

---

> ### Author Response · Authors · 2023-11-12
>
> We thank the reviewer for the comments. Below, we provide detailed responses to each point raised.
>
> **Weakness 1: it is hard to validate such explanation.**
>
> Response: We acknowledge the challenge of validating the theoretical explanation in real-world empirical studies. However, we present two key points supporting the validity of our explanation. Firstly, the HER (Hindsight Experience Replay) algorithm, a successful practical implementation, shares similarities with Algorithm 1. HER enhances exploration by relabeling sampled trajectories from other tasks, suggesting that multitask RL indeed improves the sample efficiency of myopic exploration. Secondly, drawing on the concept of diverse tasks in Linear MDP cases, we establish a connection to the Bipedal Walker environment, demonstrating that a diversity notion favoring a full rank structure of the covariance matrix of the embeddings is pivotal (as discussed in Section 6). Based on these observations, we firmly believe that our paper offers a valid perspective on explaining the success of myopic exploration.
>
> **Weakness 2: The intuition of the proof is that, if we have a base policy class with good coverage, we are able to find the optimal policy by combining the base policies with naive exploration. However, it has been well known that exploration is simple when we have good coverage. Therefore, their contribution seems not novel enough.**
>
> Response: We appreciate the reviewer's comment and we would like to have some extra clarification on the coverage of a policy class. Typically, coverage assumptions are articulated for the behavior policy of an offline dataset, often with an all-policy or single-policy concentrability assumption (refer to [1]). While constructing a diverse task set for a discrete policy class with each policy representing the optimal policy of a task is straightforward, the challenge arises when the policy set is infinite. Constructing an epsilon-covering of the policy set may lead to an exponentially large task set. However, we demonstrate that for tabular and linear cases, a diverse task set can be constructed with only polynomial (SAH) or polynomial (dH) tasks, a departure from existing literature results.
>
> **Question 1: What does Def 3 mean in linear MDP?**
>
> Response: For a detailed understanding of Def 3 in linear MDP, we refer the reviewer to Lemma 8 in Appendix E.1, which states that the coverage condition in Def 3 could be expressed as a full rank condition of the feature covariance matrix. Low MEG in Def 3 in linear MDP intuitively implies that we should construct the task set “diverse” enough such that for all $h \in [H]$, the feature covariance matrix $\mathbb{E}_{\pi} \phi_h(s_h, a_h) \phi_h(s_h, a_h)^\top$ obtained by running the current exploration policy (mixture of epsilon greedy policies) is full rank.
>
> **Question 2: Can you provide an example where Def 3 holds and the coverage of the base policies is poor?**
>
> Response: Could the reviewer elaborate on "base policies"? Def 3 is not a condition. However, if the concern is whether low MEG holds even when the current greedy policies are non-optimal, we direct the reviewer to Figure 1, where low MEG consistently holds despite the non-optimality of the current greedy policies.
>
> We hope that these responses effectively address the reviewer's comments and concerns.

---

> > ### Comment · Area_Chair_1vhQ · 2023-11-22
> > **Please respond to the author reply**
> >
> > Dear reviewer, please do respond to the author reply and let them know if this has answered your questions/concerns.

---

### Meta-Review · Area_Chair_1vhQ · 2023-12-05

**Metareview:**

(a) this paper argues that simple myopic exploration strategies can be very effective in multi-task RL settings. They argue that when there is a multi-task setting and there is ``policy sharing" across the MDPs, then exploration can be provably efficient, as compared to just standard single task learning. They conduct studies in a simple tabular, linear and bipedal walking environments.

(b) the insight is crisp and useful! I expect it to make an impact on both the theory and practice communities.

(c) the paper could provide a bit more intuition in places, could motivate the multi-task setting more clearly and articulate in what problems the assumptions are reasonable. The coverage assumptions that are required should be spelled out more explicitly

(d) A few more experiments scaled up could show the benefits clearly. More practically characterizing "what" tasks would help with exploration might also help. Trying it on more practical already existing MTRL benchmarks might also help.

**Justification For Why Not Higher Score:**

The paper could be written more clearly, the experiments could be more comprehensive and the problem setting could be motivated better.

**Justification For Why Not Lower Score:**

It is an interesting set of ideas both in theory and practice that the community should learn about, and will add value to design of algorithms in the future since they may no longer need very complex exploration methods.

---

### Decision · Program_Chairs · 2024-01-16

Accept (poster)